# The Meshless Analysis of Scale-Dependent Problems for Coupled Fields

**DOI:** 10.3390/ma13112527

**Published:** 2020-06-02

**Authors:** Jan Sladek, Vladimir Sladek, Pihua H. Wen

**Affiliations:** 1Institute of Construction and Architecture, Slovak Academy of Sciences, 84503 Bratislava, Slovakia; vladimir.sladek@savba.sk; 2School of Engineering and Materials Sciences, Queen Mary University of London, Mile End, London E14NS, UK; p.h.wen@qmul.ac.uk

**Keywords:** MLS approximation, gradients of strains, gradients of electric intensity vector, higher-grade heat flux

## Abstract

The meshless local Petrov–Galerkin (MLPG) method was developed to analyze 2D problems for flexoelectricity and higher-grade thermoelectricity. Both problems were multiphysical and scale-dependent. The size effect was considered by the strain and electric field gradients in the flexoelectricity, and higher-grade heat flux in the thermoelectricity. The variational principle was applied to derive the governing equations within the higher-grade theory of considered continuous media. The order of derivatives in the governing equations was higher than in their counterparts in classical theory. In the numerical treatment, the coupled governing partial differential equations (PDE) were satisfied in a local weak-form on small fictitious subdomains with a simple test function. Physical fields were approximated by the moving least-squares (MLS) scheme. Applying the spatial approximations in local integral equations and to boundary conditions, a system of algebraic equations was obtained for the nodal unknowns.

## 1. Introduction

In nanocomposites, nanosized particles are incorporated into a matrix. In these materials, the of surface-to-volume ratio is significantly larger than in their bulk-sized equivalents. Their properties can be improved many times with respect to those known for individual constituents. Mechanical strength, toughness and electrical or thermal conductivity can be dramatically improved in nanocomposites. Therefore, they are starting to be intensively utilized in many engineering fields. However, size-effect phenomena are observed in nanosized structures if the characteristic length of material structure is comparable with the size of the analyzed body [1,2,3,4,5]. Even some new phenomena are observed in nanosized structures, e.g., the electric polarization in centro-symmetric crystals. This is explained by the direct flexoelectricity effect [6,7,8]. In contrast to piezoelectricity, this can be viewed as a higher order effect [9]. If stresses are proportional to the gradients of the electric intensity vector, we are talking about converse flexoelectricity [10,11,12]. Strain and electric intensity vector gradients are very large in nanosized dielectrics, and they should be considered in constitutive equations.

Nanotechnology is also being successfully utilized to improve thermoelectric properties [13]. Thermoelectric materials have the potential to convert waste heat directly into electricity if the thermal conductivity is reduced without affecting the high electrical conductivity [14]. Thermal conductivity is reduced significantly in nanostructures only, as scattering of phonons is observed only on interfaces of nanostructures. Due to this phonon-scattering, thermal conductivity is reduced. Since the electrons are smaller, they are not scattered and the electric conductivity is not reduced. This requires developing a theory for heat conduction where size effects are considered.

Microstructural material characteristics are not considered in the classical continuum theory; results are size-independent. Atomistic models should be able to consider the size-effect in structural elements. However, extremely high requirements are put on computer memory in such approach. Moreover, if we are interested in global response of macro-structural elements, it is more convenient to work with advanced continuum models, in which intrinsic length-scale parameters (characteristic of material microstructure) are considered [15,16,17]. Earlier gradient theory is very complicated, due to many additional material parameters that are unavailable. Aifantis [18] simplified the former theory by introducing only one length-scale parameter. The nonlocality should be considered in the heat conduction problems if the temperature gradients are large [19]. For a special weight function in nonlocal integral expression of the heat flux vector, it is possible to transform the integral form of constitutive law into a differential relationship with a characteristic length-scale parameter representing the nonlocality. In both flexoelectric and higher-grade thermoelectric problems, we have constitutive equations with the intrinsic material parameter representing microstructure and higher derivatives of physical fields than in corresponding problems described by classical theory. Because of coupling among various physical fields, both these problems are multiphysical.

Many discontinuities and damage problems are described by nonlocal models based on the peridynamic theory [20]. Constitutive equations are written in integral form rather than partial differential equations. Various meshless methods have been successfully applied for these problems. Silling and Askari [21] applied the finite point method (FPM) to peridynamics. Moreover, there are also trends of active studies to design meshless methods that take the advantages of the classical and nonlocal theories simultaneously in a meshless framework [22,23,24].

It is necessary to have a reliable and accurate computational tool for solving the multiphysical problems described by gradient theories. Because of higher-order derivatives in governing equations, the C^1^-continuous elements are required in numerical domain discretization methods in order to guarantee the continuity of field variables and their derivatives on interfaces of elements. It is a difficult task to obey such a requirement, though certain special elements were developed [25,26]. It is more convenient to develop the mixed formulation in the finite element method (FEM) [20,21,25,26]. Both the primary field and its derivatives are approximated as independent field variables in the mixed FEM with using C^0^ continuous elements. However, numerous degrees of freedom (DOFs) are used in each element, which make it prohibitively expensive for practical use. The order of continuity of the moving least-square (MLS) approximation in the meshless local Petrov–Galerkin method (MLPG) can be tuned to a desired value very easily [22,23,24,27,28,29]. Therefore, it is a natural ambition to apply the MLPG with MLS approximation to problems described by gradient theories with higher-order derivatives in governing equations.

In the present study, the authors have developed a meshless method based on the MLPG weak-formulation for numerical solution of multiphysical problems in dielectric and thermoelectric solids. Both the direct and converse flexoelectricity is considered in dielectric solids. Nodal points are spread on the analyzed domain and each node is surrounded by a small circular subdomain for simplicity, but without loss of generality, for consideration of governing equations in local weak sense within the MLPG method for solution. The spatial variations of primary physical fields were approximated by the moving least-squares (MLS) scheme. After performing the spatial integrations, a system of algebraic or ordinary differential equations for unknown nodal values was obtained. The essential boundary conditions on the global boundary were satisfied in strong sense by the collocation of approximated fields at nodes where essential boundary conditions were prescribed. Illustrative numerical examples are presented and discussed with focusing on comparison of results obtained by the gradient theory with those obtained by classical theory.

## 2. The MLPG for Flexoelectricity

### 2.1. The Direct and Converse Flexoelectricity

The electric enthalpy density for piezoelectric solids with strain and electric intensity vector gradients can be written as [25,26,27,28,29,30,31]
$$H=\frac{1}{2}c_{ijkl}\varepsilon_{ij}\varepsilon_{kl}-\frac{1}{2}a_{ij}E_{i}E_{j}-e_{kji}\varepsilon_{ij}E_{k}+\frac{1}{2}g_{jklmni}\eta_{jkl}\eta_{mni}-f_{ijkl}E_{i}\eta_{jkl}+$$
(1)$$+b_{klij}\varepsilon_{ij}E_{k,l}-\frac{1}{2}h_{ijkl}E_{i,j}E_{k,l}$$
where *a* and *c* denote the second-order permittivity and the fourth-order elastic constant tensors, respectively. Symbols *e* and *f* are used for the piezoelectric and flexoelectric coefficients, respectively. Symbol *g* denotes the higher order elastic coefficients. Finally, the symbols *b* and *h* are used for the converse flexoelectric coefficients and higher-order electric parameters, respectively.

The strain tensor $\varepsilon_{ij}$ and the electric field vector $E_{j}$ are related to the displacements $u_{i}$ and the electric potential ϕ by
(2)$$\varepsilon_{ij}=\frac{1}{2}\left(u_{i,j}+u_{j,i}\right),\;E_{j}=-\phi_{,j}$$

The strain-gradients are denoted by symbol *η*
(3)$$\eta_{ijk}=\varepsilon_{ij,k}=\frac{1}{2}\left(u_{i,jk}+u_{j,ik}\right).$$

The constitutive equations are obtained from the electric enthalpy density expression (1) as
$$\sigma_{ij}=\frac{\partial H}{\partial \varepsilon_{ij}}=c_{ijkl}\varepsilon_{kl}-e_{kij}E_{k}-b_{klij}E_{k,l}$$
$$\tau_{jkl}=\frac{\partial H}{\partial \eta_{jkl}}=-f_{ijkl}E_{i}+g_{jklmni}\eta_{nmi}$$
(4)$$D_{i}=-\frac{\partial H}{\partial E_{i}}=a_{ij}E_{j}+e_{ijk}\varepsilon_{jk}+f_{ijkl}\eta_{jkl},\;$$
$$Q_{ij}=-\frac{\partial H}{\partial E_{i,j}}=b_{ijkl}\varepsilon_{kl}+h_{ijkl}E_{k,l}$$
where $\sigma_{ij}$, $D_{i}$, $\tau_{jkl}$ and $Q_{ij}$ are the stress tensor, electric displacements, higher order stress and electric quadrupole, respectively.

In the simplified gradient theory [18,32,33] only one internal length-scale parameter *l* is present. Then, the higher-order elastic parameters $g_{jklmni}$ are proportional to the conventional elastic stiffness coefficients $c_{klmn}$ and the length material parameter *l* [34,35] as
(5)$$g_{jklmni}=l^{2}c_{jkmn}\delta_{li},$$
with $\delta_{li}$ being the Kronecker delta.

A similar idea of simplification has been applied to the higher-order electric parameters
(6)$$h_{ijkl}=q^{2}a_{ik}\delta_{jl},$$
where *q* is another length-scale parameter.

In the simplified direct flexoelectricity there are considered two independent parameters $f_{1}$ and $f_{2}$ for the direct flexoelectric coefficient $f_{ijkl}$, $f_{ijkl}=f_{1}\delta_{jk}\delta_{il}+f_{2}\left(\delta_{ij}\delta_{kl}+\delta_{ik}\delta_{jl}\right)$ [25]. Taking into account the above simplifications, the electric enthalpy density can be written as
$$H=\frac{1}{2}c_{ijkl}\varepsilon_{ij}\varepsilon_{kl}-\frac{1}{2}a_{ij}E_{i}E_{j}-e_{kji}\varepsilon_{ij}E_{k}+\frac{l^{2}}{2}c_{jkmn}\eta_{jkl}\eta_{mnl}-f_{1}E_{i}\eta_{kki}-f_{2}E_{i}\left(\eta_{ikk}+\eta_{jij}\right)+$$
(7)$$+b_{klij}\varepsilon_{ij}E_{k,l}-\frac{q^{2}}{2}a_{ik}E_{i,j}E_{k,j}.$$

Finally, we reduce the number of independent converse flexoelectric coefficients $b_{ijkl}$. The stresses induced by electric intensity vector in the orthotropic piezoelectric material with poling direction along the transversal isotropy $x_{3}$-axis can be written as
$$\sigma_{11}=e_{31}E_{3},\;\sigma_{33}=e_{33}E_{3},\;\sigma_{13}=e_{15}E_{1},$$
with
(8)$$e_{kij}=\left(e_{31}\delta_{i1}\delta_{j1}+e_{33}\delta_{i3}\delta_{j3}\right)\delta_{k3}+e_{15}\left(\delta_{i1}\delta_{j3}+\delta_{i3}\delta_{j1}\right)\delta_{k1}$$
where standard Voight notation is applied [36].

A similar form is considered for induced stresses induced by the converse flexoelectricity
(9)$$\sigma_{ij}=\delta_{ij}b_{1}\left(E_{1,1}+E_{3,3}\right),\;\sigma_{13}=\sigma_{31}=b_{2}E_{1,3}+b_{3}E_{3,1}$$
with three independent converse flexoelectric coefficients $b_{1}$, $b_{2}$ and $b_{3}$ by $b_{klij}=b_{1}\delta_{ij}\delta_{kl}+\left(\delta_{i1}\delta_{j3}+\delta_{i3}\delta_{j1}\right)\left(b_{2}\delta_{k1}\delta_{l3}+b_{3}\delta_{k3}\delta_{l1}\right)$. Note that $b_{1}$, $b_{2}$ and $b_{3}$ are three independent material coefficients, but are not components of a vector, similar as $f_{1}$, $f_{2}$ in case of direct flexoelectricity.

Then, the final form of the electric enthalpy is given by
(10)$$\begin{array}{ll} H & =\frac{1}{2}c_{ijkl}\varepsilon_{ij}\varepsilon_{kl}-\frac{1}{2}a_{ij}E_{i}E_{j}-e_{31}\varepsilon_{11}E_{3}-e_{33}\varepsilon_{33}E_{3}-e_{15}\left(\varepsilon_{13}+\varepsilon_{31}\right)E_{1}+\frac{l^{2}}{2}c_{jkmn}\eta_{jkl}\eta_{mnl}-\\ & -f_{1}E_{i}\eta_{kki}-f_{2}E_{i}\left(\eta_{ikk}+\eta_{jij}\right)+b_{1}\varepsilon_{kk}E_{i,i}+\left(b_{2}E_{1,3}+b_{3}E_{3,1}\right)\left(\varepsilon_{13}+\varepsilon_{31}\right)-\frac{q^{2}}{2}a_{ik}E_{i,j}E_{k,j} \end{array}$$

The constitutive Equations (4) for orthotropic materials in 2D problem (considered in $\left(x_{1}x_{3}\right)$-plane) can be rewritten into a matrix form as
$$\left[\begin{array}{c} \sigma_{11}\\ \sigma_{33}\\ \sigma_{13} \end{array}\right]=\left[\begin{array}{ccc} c_{11} & c_{13} & 0\\ c_{13} & c_{33} & 0\\ 0 & 0 & c_{44} \end{array}\right]\left[\begin{array}{c} \varepsilon_{11}\\ \varepsilon_{33}\\ 2\varepsilon_{13} \end{array}\right]-\left[\begin{array}{c} \begin{array}{cc} 0 & e_{31} \end{array}\\ \begin{array}{cc} 0 & e_{33} \end{array}\\ \begin{array}{cc} e_{15} & 0 \end{array} \end{array}\right]\left[\begin{array}{c} E_{1}\\ E_{3} \end{array}\right]-\left[\begin{array}{c} \begin{array}{cc} b_{1} & 0 \end{array}\\ \begin{array}{cc} b_{1} & 0 \end{array}\\ \begin{array}{cc} 0 & b_{3} \end{array} \end{array}\right]\left[\begin{array}{c} E_{1,1}\\ E_{3,1} \end{array}\right]-\left[\begin{array}{c} \begin{array}{cc} 0 & b_{1} \end{array}\\ \begin{array}{cc} 0 & b_{1} \end{array}\\ \begin{array}{cc} b_{2} & 0 \end{array} \end{array}\right]\left[\begin{array}{c} E_{1,3}\\ E_{3,3} \end{array}\right]=$$
(11)$$=C\left[\begin{array}{c} \varepsilon_{11}\\ \varepsilon_{33}\\ 2\varepsilon_{13} \end{array}\right]-\Lambda \left[\begin{array}{c} E_{1}\\ E_{3} \end{array}\right]-\Phi_{1}\left[\begin{array}{c} E_{1,1}\\ E_{3,1} \end{array}\right]-\Phi_{3}\left[\begin{array}{c} E_{1,3}\\ E_{3,3} \end{array}\right],$$
$$\left[\begin{array}{c} D_{1}\\ D_{3} \end{array}\right]=\left[\begin{array}{ccc} \begin{array}{c} 0\\ e_{31} \end{array} & \begin{array}{c} 0\\ e_{33} \end{array} & \begin{array}{c} e_{15}\\ 0 \end{array} \end{array}\right]\left[\begin{array}{c} \varepsilon_{11}\\ \varepsilon_{33}\\ 2\varepsilon_{13} \end{array}\right]+\left[\begin{array}{cc} a_{1} & 0\\ 0 & a_{2} \end{array}\right]\left[\begin{array}{c} E_{1}\\ E_{3} \end{array}\right]+$$
(12)$$\begin{array}{l} +\left[\begin{array}{ccc} f_{1}+2f_{2} & f_{1} & 0\\ 0 & 0 & f_{2} \end{array}\right]\left[\begin{array}{c} \varepsilon_{11,1}\\ \varepsilon_{33,1}\\ 2\varepsilon_{13,1} \end{array}\right]+\left[\begin{array}{ccc} 0 & 0 & f_{2}\\ f_{1} & f_{1}+2f_{2} & 0 \end{array}\right]\left[\begin{array}{c} \varepsilon_{11,3}\\ \varepsilon_{33,3}\\ 2\varepsilon_{13,3} \end{array}\right]=\\ =\Lambda^{T}\left[\begin{array}{c} \varepsilon_{11}\\ \varepsilon_{33}\\ 2\varepsilon_{13} \end{array}\right]+A\left[\begin{array}{c} E_{1}\\ E_{3} \end{array}\right]+F_{1}\left[\begin{array}{c} \varepsilon_{11,1}\\ \varepsilon_{33,1}\\ 2\varepsilon_{13,1} \end{array}\right]+F_{3}\left[\begin{array}{c} \varepsilon_{11,3}\\ \varepsilon_{33,3}\\ 2\varepsilon_{13,3} \end{array}\right] \end{array}$$
(13)$$\left[\begin{array}{c} \tau_{11k}\\ \tau_{33k}\\ \tau_{13k} \end{array}\right]=-F_{k}^{T}\left[\begin{array}{c} E_{1}\\ E_{3} \end{array}\right]+l^{2}C\left[\begin{array}{c} \varepsilon_{11,k}\\ \varepsilon_{33,k}\\ 2\varepsilon_{13,k} \end{array}\right],$$
(14)$$\left[\begin{array}{c} Q_{1k}\\ Q_{3k} \end{array}\right]=\Phi_{k}^{T}\left[\begin{array}{c} \varepsilon_{11}\\ \varepsilon_{33}\\ 2\varepsilon_{13} \end{array}\right]+q^{2}A\left[\begin{array}{c} E_{1,k}\\ E_{3,k} \end{array}\right].$$

The Voigt notation is applied for piezoelectric and dielectric coefficients, similarly to elastic coefficients as commonly used in literature [36].

Using the variational principle of least action, it is possible to derive the governing equations for the considered constitutive equations [37]
$$\sigma_{ij,j}(x)-\tau_{ijk,jk}(x)=0,$$
(15)$$D_{i,i}(x)-Q_{ij,ji}(x)=0.$$

Essential and natural boundary conditions (b.c.) follow from the variational formulation of boundary value problems:(1)Essential b.c.:$$u_{i}(x)={\bar{u}}_{i}(x)\;\mathrm{on}\;\Gamma_{u},\;\Gamma_{s}\subset \Gamma$$
$$s_{i}(x)={\bar{s}}_{i}\;\mathrm{on}\;\Gamma_{s},\;\Gamma_{s}\subset \Gamma$$
(16)$$\phi (x)=\bar{\phi }(x)\;\mathrm{on}\;\Gamma_{\phi },\;\Gamma_{\phi }\subset \Gamma$$
$$p(x)=\frac{\partial \phi }{\partial n}=\bar{p}(x)\;\mathrm{on}\;\Gamma_{p},\;\Gamma_{p}\subset \Gamma$$(2)Natural b.c.:
$$t_{i}(x)={\bar{t}}_{i}(x)\;\mathrm{on}\;\Gamma_{t},\;\Gamma_{t}\cup \Gamma_{u}=\Gamma ,\;\Gamma_{t}\cap \Gamma_{u}=\emptyset$$
$$R_{i}(x)={\bar{R}}_{i}(x)\;\mathrm{on}\;\Gamma_{R},\;\Gamma_{R}\cup \Gamma_{s}=\Gamma ,\;\Gamma_{R}\cap \Gamma_{s}=\emptyset$$
(17)$$S(x)=\bar{S}(x)\;\mathrm{on}\;\Gamma_{S},\;\Gamma_{S}\cup \Gamma_{\phi }=\Gamma ,\;\Gamma_{S}\cap \Gamma_{\phi }=\emptyset$$
$$Z(x)=\bar{Z}(x)\;\mathrm{on}\;\Gamma_{Z},\;\Gamma_{Z}\cup \Gamma_{p}=\Gamma ,\;\Gamma_{Z}\cap \Gamma_{p}=\emptyset ,$$
where
(18)$$s_{i}:=\frac{\partial u_{i}}{\partial n},\;p:=\frac{\partial \phi }{\partial n},\;R_{i}:=n_{k}n_{j}\tau_{ijk},\;Z:=n_{i}n_{j}Q_{ij},$$
and the traction vector and the electric charge are defined as
(19)$$t_{i}:=n_{j}\left(\sigma_{ij}-\tau_{ijk,k}\right)-\frac{\partial \rho_{i}}{\partial \pi }+\sum_{c}\left\|\rho_{i}(x^{c})\right\|\delta (x-x^{c}),$$
(20)$$S:=n_{k}\left(D_{k}-Q_{kj,j}\right)-\frac{\partial \alpha }{\partial \pi }+\sum_{c}\left\|\alpha (x^{c})\right\|\delta (x-x^{c}),$$
with $\rho_{i}:=n_{k}\pi_{j}\tau_{ijk}$, $\alpha :=n_{i}\pi_{j}Q_{ij}$, *δ(**x**)* being the Dirac delta function and $\pi_{i}$ is the Cartesian component of the unit tangent vector on Γ.

The jump at a corner (**x**^c^) on the oriented boundary contour Γ is defined as
(21)$$\left\|\rho_{i}(x^{c})\right\|:=\rho_{i}(x^{c}-)-\rho_{i}(x^{c}+),$$
(22)$$\left\|\alpha (x^{c})\right\|:=\alpha (x^{c}-)-\alpha (x^{c}+).$$

### 2.2. The MLPG Formulation

The presence of gradients of strains and electric intensity vector in the electric enthalpy requires C^1^ continuous interpolations of primary fields, i.e., displacements and electric potential. Recently, the mixed FEM was developed for considered electro-elasticity problem [38]. The meshless local Petrov–Galerkin method (MLPG) with the moving least-square (MLS) approximation is convenient approach for problems with higher-order derivatives, since the order of continuity can be tuned to a desired value [27,28,29].

The MLPG method is based on the local weak-form with local fictitious subdomains $\Omega^{q}$ constructed for the node $x^{q}$ which is either interior node $x^{i}\in \Omega$ or boundary node $x^{b}\in \partial \Omega =\Gamma$ at which natural boundary conditions are prescribed (see Figure 1). The geometry of this subdomain can be arbitrary. However, it is appropriate to select a circular shape for simple numerical evaluation of integrals. One can write the local weak-form of the first governing Equation (15) as
(23)$$\int_{\Omega^{q}}\left[\sigma_{ij,j}(x)-\tau_{ijk,jk}(x)\right]\;u_{im}^{*}(x)\;d\Omega =0$$
where $u_{im}^{*}(x)$ is a test function.

Applying the Gauss divergence theorem to domain integrals in (23) one can write
(24)$$\int_{\partial \Omega^{q}}\left[\sigma_{ij}(x)-\tau_{ijk,k}(x)\right]n_{j}(x)u_{im}^{*}(x)d\Gamma -\int_{\Omega^{q}}\left[\sigma_{ij}(x)-\tau_{ijk,k}(x)\right]u_{im,j}^{*}(x)d\Omega =0,$$
where $\partial \Omega^{q}$ is the boundary of the local subdomain which consists of three parts $\partial \Omega^{q}=L^{q}\cup \Gamma_{t}^{q}\cup \Gamma_{u}^{q}$, in general, since $\partial \Omega^{q}=\left(\partial \Omega^{q}\cap \Omega \right)\cup \left(\partial \Omega^{q}\cap \Gamma \right)$ and $L^{q}:=\partial \Omega^{q}\cap \Omega$, $\Gamma^{q}:=\partial \Omega^{q}\cap \Gamma =\left(\partial \Omega^{q}\cap \Gamma_{t}\right)\cup \left(\partial \Omega^{q}\cap \Gamma_{u}\right)=\Gamma_{t}^{q}\cup \Gamma_{u}^{q}$.

If a Heaviside step function is chosen for the test function $u_{ik}^{*}(x)$ in each subdomain as
(25)$$u_{ik}^{*}(x)=\left\{\begin{array}{c} \delta_{ik}\;\mathrm{at}\;x\in \Omega^{q}\cup \partial \Omega^{q}\\ 0\;\mathrm{at}\;x\notin \Omega^{q}\cup \partial \Omega^{q} \end{array}\right.,$$
the local weak-form (24) is transformed into the local integral equation
(26)$$\int_{L^{q}+\Gamma_{u}^{q}}n_{j}\left(\sigma_{ij}-\tau_{ijk,k}\right)d\Gamma +\rho_{i}(x_{t}^{f})-\rho_{i}(x_{t}^{s})=-\int_{\Gamma_{t}^{q}}{\bar{t}}_{i}d\Gamma ,$$
where we have utilized the fact ${\left.n_{j}\left(\sigma_{ij}-\tau_{ijk,k}\right)\right|}_{\Gamma_{t}^{q}}={\left.{\bar{t}}_{i}\right|}_{\Gamma_{t}^{q}}+{\left.\left(\frac{\partial \rho_{i}}{\partial \pi }-\sum_{c}\left\|\rho_{i}(x^{c})\right\|\delta (x-x^{c})\right)\right|}_{\Gamma_{t}^{q}}$ and
$$\int_{\Gamma_{t}^{q}}\left(\frac{\partial \rho_{i}}{\partial \pi }-\sum_{c}\left\|\rho_{i}(x^{c})\right\|\delta (x-x^{c})\right)d\Gamma =\rho_{i}(x_{t}^{f})-\rho_{i}(x_{t}^{s})$$
with $x_{t}^{f}$, $x_{t}^{s}$ standing for the final and starting points on $\Gamma_{t}^{q}$.

Similarly, we get local integral equation for the second governing Equation (15)
(27)$$\int_{L^{q}+\Gamma_{\phi }^{q}}n_{j}\left(D_{j}-Q_{ij,i}\right)d\Gamma +\alpha (x_{S}^{f})-\alpha (x_{S}^{s})=-\int_{\Gamma_{S}^{q}}\bar{S}d\Gamma .$$

For numerical solution of the above integral Equations (26) and (27), the MLS approximation of trial functions is applied. The primary fields (mechanical displacements and electric potential) are given by [28]
(28)$$u^{h}(x)=N^{T}(x)\cdot \hat{u}=\sum_{a=1}^{n}N^{a}(x){\hat{u}}^{a},\;\;\phi^{h}(x)=\sum_{a=1}^{n}N^{a}(x){\hat{\phi }}^{a},$$
where ${\hat{u}}^{a}={\left({\hat{u}}_{1}^{a},\;{\hat{u}}_{3}^{a}\right)}^{T}$ and ${\hat{\phi }}^{a}$ are fictitious nodal parameters for displacements and electric potential, respectively and $N^{a}(x)$ is the shape function related to the node *a*. Recall that the shape functions do not satisfy the Kronecker δ-property. Therefore the nodal unknowns ${\hat{u}}^{a}$, ${\hat{\phi }}^{a}$ are not nodal values of displacements and electric potential, but these nodal unknowns can be used for approximation of real nodal values at an arbitrary nodal point $x^{q}$ by utilizing (28) at $x=x^{q}$. The number of nodes, *n*, used for the approximation is determined by the size of support domain of the weight function $w^{a}(x)$ of each node $x^{a}$. The node $x^{a}$ is involved into the approximation at the sample point x, only if x lies within the support domain of the node $x^{a}$, i.e., if $w^{a}(x)\neq 0$. A necessary condition for a regular MLS approximation is that at least m weight functions are non-zero (i.e., n≥m) for each sample point x∈Ω, where m is the order of the complete basis functions used in MLS approximation. A sufficient number of nodes must be involved in order to ensure the regularity of evaluation of shape functions [27]. A small size of the support domains may induce larger oscillations in the nodal shape functions. In standard discretization methods like the FEM, there are observed discontinuities on interfaces of elements if continuity of approximation is not sufficient. In meshless methods there are no element interfaces and we can use the forth-order spline-type weight function with $C^{1}-$ continuity even for our problem with derivatives of the third order. For the MLS approximation, we have used the complete set of monomials of the 3rd order (m=10) [27,28,29] and the weight function has the following form
(29)$$w^{a}(x)=\left\{\begin{array}{ll} 1-6{\left(\frac{d^{a}}{r^{a}}\right)}^{2}+8{\left(\frac{d^{a}}{r^{a}}\right)}^{3}-3{\left(\frac{d^{a}}{r^{a}}\right)}^{4}, & 0\leq d^{a}\leq r^{a}\\ 0, & d^{a}\geq r^{a} \end{array}\right.,$$
where $d^{a}=\left\|x-x^{a}\right\|$ and $r^{a}$ is the size of the support domain. In numerical examples, we have used the numerical model verified in plenty of numerical experiments (e.g., [27,28,39,40]): the radius of local subdomain $\rho^{q}=0.5\delta$, radius of support domain $r^{a}=3.001\delta$, where δ is the minimal distance of any two nodes. If the number of nodes supporting the approximation at a sample point x is smaller than 15 (n<15), additional closest nodes are supplemented as supporting nodes in the adopted numerical model. In problems with simple geometry, it is proposed to employ regular distribution of nodes. The circular shape of subdomain enables us to utilize polar coordinates and facilitate the integrations. Since the radius of local subdomain is rather small, the subdomain $\Omega^{b}$ around a boundary node can be considered as a section of complete circular domain and $L^{b}$ is a circular arc (Figure 1). The Gaussian quadrature is used for integrations with using polar coordinate system.

The strains and electric intensity vector are approximated by
$$\varepsilon^{h}(x)=\left[\begin{array}{c} \varepsilon_{11}^{h}\\ \varepsilon_{33}^{h}\\ 2\varepsilon_{13}^{h} \end{array}\right]=\sum_{a=1}^{n}B^{a}(x){\hat{u}}^{a},\;E^{h}(x)=\left[\begin{array}{c} E_{1}^{h}\\ E_{3}^{h} \end{array}\right]=-\sum_{a=1}^{n}P^{a}(x){\hat{\phi }}^{a},$$
(30)$$\varepsilon_{,k}^{h}(x)=\left[\begin{array}{c} \varepsilon_{11,k}^{h}\\ \varepsilon_{33,k}^{h}\\ 2\varepsilon_{13,k}^{h} \end{array}\right]=\sum_{a=1}^{n}B_{,k}^{a}(x){\hat{u}}^{a},\;E_{,k}^{h}(x)=\left[\begin{array}{c} E_{1,k}^{h}\\ E_{3,k}^{h} \end{array}\right]=-\sum_{a=1}^{n}P_{,k}^{a}(x){\hat{\phi }}^{a}$$
where
(31)$$B^{a}(x)=\left[\begin{array}{cc} \begin{array}{c} N_{,1}^{a}\\ 0\\ N_{,3}^{a} \end{array} & \begin{array}{c} 0\\ N_{,3}^{a}\\ N_{,1}^{a} \end{array} \end{array}\right],\;P^{a}(x)=\left[\begin{array}{c} N_{,1}^{a}\\ N_{,3}^{a} \end{array}\right],\;B_{,k}^{a}(x)=\left[\begin{array}{cc} \begin{array}{c} N_{,1k}^{a}\\ 0\\ N_{,3k}^{a} \end{array} & \begin{array}{c} 0\\ N_{,3k}^{a}\\ N_{,1k}^{a} \end{array} \end{array}\right],\;P_{,k}^{a}(x)=\left[\begin{array}{c} N_{,1k}^{a}\\ N_{,3k}^{a} \end{array}\right].$$

The first part of the traction vector (19), ${\tilde{t}}_{i}(x)=n_{j}(\sigma_{ij}-\tau_{ijk,k})$, can be approximated at a boundary point $x\in \partial \Omega^{b}$ in terms of primary fields as
(32)$${\tilde{t}}^{h}(x)=N(x)C\sum_{a=1}^{n}\left(B^{a}(x)-l^{2}B_{,kk}^{a}(x)\right){\hat{u}}^{a}+N(x)\sum_{a=1}^{n}\left(\Lambda P^{a}(x)+\Phi_{k}P_{,k}^{a}(x)-F_{k}^{T}P_{,k}^{a}(x)\right){\hat{\phi }}^{a},$$
where the matrices $C,\Lambda ,\Phi_{k},F_{k}^{T}$ are defined in Equations (11) and (13) and the matrix N(x) is related to the normal vector n(x) on $\partial \Omega^{b}$ by
(33)$$N(x)=\left[\begin{array}{c} \begin{array}{ccc} n_{1} & 0 & n_{3} \end{array}\\ \begin{array}{ccc} 0 & n_{3} & n_{1} \end{array} \end{array}\right]$$
Again, the first part of the electric charge (27), $\tilde{S}(x)=n_{j}(D_{j}-Q_{ij,i})$, has to be approximated. In the first step the expression of $Q_{ij,i}$ is given by
$$n_{j}Q_{ij,i}=n_{j}\left[\begin{array}{cc} \partial_{1} & \partial_{3} \end{array}\right]\left[\begin{array}{c} Q_{1j}\\ Q_{3j} \end{array}\right]=n_{j}\left[\begin{array}{cc} \partial_{1} & \partial_{3} \end{array}\right]\left\{\Phi_{j}^{T}\sum_{a=1}^{n}B^{a}(x){\hat{u}}^{a}-q^{2}A\sum_{a=1}^{n}P_{,j}^{a}(x){\hat{\phi }}^{a}\right\}=$$
(34)$$=n_{j}(x)\left\{\sum_{a=1}^{n}\Psi_{j}^{a}(x){\hat{u}}^{a}-q^{2}\sum_{a=1}^{n}\Pi_{j}^{a}(x){\hat{\phi }}^{a}\right\},$$
where,
$$\Psi_{1}^{aT}(x)=\left[\begin{array}{c} b_{1}N_{,11}^{a}+b_{3}N_{,33}^{a}\\ (b_{1}+b_{3})N_{,13}^{a} \end{array}\right],\;\Psi_{3}^{aT}(x)=\left[\begin{array}{c} (b_{1}+b_{2})N_{,13}^{a}\\ b_{2}N_{,11}^{a}+b_{1}N_{,33}^{a} \end{array}\right],\;\Pi_{j}^{a}(x)=\left[\begin{array}{c} a_{1}N_{,11j}^{a}\\ a_{2}N_{,33j}^{a} \end{array}\right].$$

Now, $\tilde{S}(x)$ can be approximated by
(35)$$\begin{array}{ll} {\tilde{S}}^{h}(x) & =N(x)\left\{\sum_{a=1}^{n}\left(\Lambda^{T}B^{a}(x)+F_{k}B_{,k}^{a}(x)\right){\hat{u}}^{a}-A\sum_{a=1}^{n}P^{a}(x){\hat{\phi }}^{a}\right\}-\\ & n_{j}(x)\left\{\sum_{a=1}^{n}\Psi_{j}^{a}(x){\hat{u}}^{a}-q^{2}\sum_{a=1}^{n}\Pi_{j}^{a}(x){\hat{\phi }}^{a}\right\} \end{array}$$
with $N(x)=\left[\begin{array}{cc} n_{1}(x) & n_{3}(x) \end{array}\right]$.

The essential boundary conditions are satisfied in the strong-form at nodal points $\zeta^{b}\in \left(\Gamma_{u}^{b}\cup \Gamma_{s}^{b}\cup \Gamma_{\phi }^{b}\cup \Gamma_{p}^{b}\right)\subset \partial \Omega$. If the approximation formulas (28) and (30) are used one can write
(36)$$\begin{array}{cc} \; & \;\sum_{a=1}^{n}N^{a}\left(\zeta^{b}\right){\hat{u}}^{a}=\bar{u}\left(\zeta^{b}\right)\;\;\;\;\;\;\;\;\mathrm{for}\;\;\;\zeta^{b}\in \;\Gamma_{u}^{b},\;\;\;\;\;\;\;\;\;\;\;\;\;N\left(\zeta^{b}\right)\sum_{a=1}^{n}P^{a}\left(\zeta^{b}\right){\hat{u}}^{a}=\bar{s}\left(\zeta^{b}\right)\;\;\;\;\;\;\;\;\mathrm{for}\;\;\;\zeta^{b}\in \;\Gamma_{s}^{b}\\ \; & \;\sum_{a=1}^{n}N^{a}\left(\zeta^{b}\right){\hat{\phi }}^{a}=\bar{\phi }\left(\zeta^{b}\right)\;\;\;\;\;\;\;\;\mathrm{for}\;\;\;\zeta^{b}\in \;\Gamma_{\phi }^{b},\;\;\;\;\;\;\;\;\;\;\;\;\;N\left(\zeta^{b}\right)\sum_{a=1}^{n}P^{a}\left(\zeta^{b}\right){\hat{\phi }}^{a}=\bar{p}\left(\zeta^{b}\right)\;\;\;\;\;\;\;\;\mathrm{for}\;\;\;\zeta^{b}\in \;\Gamma_{p}^{b} \end{array}$$

Substituting the MLS-approximation (32) and (35) into the local boundary-domain integral Equations (26) and (27), we obtain the system of algebraic equations for unknown nodal quantities
(37)$$\begin{array}{cc} \; & \int_{L^{q}+\Gamma_{u}^{q}}N\left(x\right)C\sum_{a}\left(B^{a}\left(x\right)-l^{2}B_{,kk}^{a}\left(x\right)\right){\hat{u}}^{a}d\Gamma \left(x\right)+\\ \; & \;\;\;\;\;\;\;\;\;+\int_{L^{q}+\Gamma_{u}^{q}}N\left(x\right)\sum_{a=1}^{n}\left(\Lambda P^{a}\left(x\right)+\Phi_{k}P_{,k}^{a}\left(x\right)-F_{k}^{T}P_{,k}^{a}\left(x\right)\right){\hat{\phi }}^{a}d\Gamma \left(x\right)+\;\\ \; & \;\;\;\;\;\;\;\;\;+n_{k}\left(x_{t}^{f}\right)\Pi \left(x_{t}^{f}\right)\left(l^{2}C\sum_{a}B_{,k}^{a}\left(x_{t}^{f}\right){\hat{u}}^{a}+F_{k}^{T}\sum_{a}P^{a}\left(x_{t}^{f}\right){\hat{\phi }}^{a}\right)-\\ \; & \;\;\;\;\;\;\;\;-n_{k}\left(x_{t}^{s}\right)\Pi \left(x_{t}^{s}\right)\left(l^{2}C\sum_{a}B_{,k}^{a}\left(x_{t}^{s}\right){\hat{u}}^{a}+F_{k}^{T}\sum_{a}P^{a}\left(x_{t}^{s}\right){\hat{\phi }}^{a}\right)=-\int_{\Gamma_{t}^{q}}\bar{t}\left(x\right)d\Gamma \end{array}$$
(38)$$\begin{array}{cc} \; & \int_{L^{q}+\Gamma_{\phi }^{q}}N(x)\left[\sum_{a=1}^{n}\left(\Lambda^{T}B^{a}\left(x\right)+F_{k}B_{,k}^{a}\left(x\right)\right){\hat{u}}^{a}-A\sum_{a=1}^{n}P^{a}\left(x\right){\hat{\phi }}^{a}\right]d\Gamma \left(x\right)-\\ \; & \;\;\;\;\;\;\;\;\;\;\;\;\;\;-\int_{L^{q}+\Gamma_{\phi }^{q}}n_{j}\left(x\right)\left[\sum_{a=1}^{n}\Psi_{j}^{a}\left(x\right){\hat{u}}^{a}-q^{2}\sum_{a=1}^{n}\Pi_{j}^{a}\left(x\right){\hat{\phi }}^{a}\right]d\Gamma \left(x\right)+\\ \; & \;\;\;\;\;\;\;\;\;\;\;\;\;+N\left(x_{S}^{f}\right)\pi_{k}\left(x_{S}^{f}\right)\left(\Phi_{k}^{T}\sum_{a}B^{a}\left(x_{S}^{f}\right){\hat{u}}^{a}-q^{2}A\sum_{a}P_{,k}^{a}\left(x_{S}^{f}\right){\hat{\phi }}^{a}\right)\\ \; & \;\;\;\;\;\;\;\;\;\;\;\;-N\left(x_{S}^{s}\right)\pi_{k}\left(x_{S}^{s}\right)\left(\Phi_{k}^{T}\sum_{a}B^{a}\left(x_{S}^{s}\right){\hat{u}}^{a}-q^{2}A\sum_{a}P_{,k}^{a}\left(x_{S}^{s}\right){\hat{\phi }}^{a}\right)=-\int_{\Gamma_{S}^{q}}\bar{S}\left(x\right)d\Gamma \end{array}$$
which are considered on the sub-domains adjacent to the interior nodes $x^{q}\in \Omega$ as well as to the boundary nodes on $x^{q}\in \Gamma_{t}^{q}\subset \partial \Omega$ and/or $x^{q}\in \Gamma_{S}^{q}\subset \partial \Omega$ with
(39)$$\Pi (x)=\left[\begin{array}{c} \begin{array}{ccc} \pi_{1}(x) & 0 & \pi_{3}(x) \end{array}\\ \begin{array}{ccc} 0 & \pi_{3}(x) & \pi_{1}(x) \end{array} \end{array}\right].$$

### 2.3. Numerical Examples

A square panel under bending in Figure 2 is analyzed by the FEM [37] and the MLPG. The piezoelectric material PZT-5H is chosen for the study.

Polarization of material is considered along $x_{3}$ coordinate. Following geometry and load are considered in numerical analysis: $w=1.0\times 10^{-7}\;m$, $t_{1}=1.0\times 10^{6}\;\mathrm{MPa}$. The size effect is control by parameter α, defined by $l^{2}=\alpha l_{0}^{2}$ with micro-length scale parameter $l_{0}=5\times 10^{-9}\;m$. The parameter α is used just in parametric study for investigation of influence of the micro-length scale parameter l on the physical response. The flexo-electric coefficients are vanishing here.

The variation of the panel deflection along $x_{1}$ is presented in Figure 3. Results are obtained by classical and gradient theories. Recall that the classical and higher-grade problems are to be solved individually, since the governing equations in the higher-grade model are given by the partial differential equations (PDE) of higher order than in classical theory and moreover, some additional boundary conditions are required. In gradient theory only strain gradients are considered in constitutive equations, while gradients of electric intensity vector are vanishing in this example. The deflections resulting from the gradient theory are reduced with respect to those obtained by the classical approach. The FEM and MLPG results are in a good agreement. In the FEM calculations, the special Argyris element [41] was employed in order to achieve higher continuity of approximated field variables.

A square plate with a central crack with the geometric parameters w=5a, $a=1.0\times 10^{-7}\;m$ is analyzed (see Figure 4) in the next example. On the top and the bottom boundaries of the plate a combined electro-mechanical loading with $t_{3}=1.17\;\mathrm{MPa}$ and $D_{3}=-5\times 10^{-4}{\;C/m}^{2}$ is applied. The crack-faces are free of mechanical tractions and electrical displacements. 

The flexoelectric coefficients are considered to be $f_{1}=f_{2}=f_{0}=1\times 10^{-8}\;C/m$. The converse flexoelectric coefficients and length scale parameter for the higher-order electric parameters are selected as $b_{1}=b_{2}=b_{3}=b_{0}=5\times 10^{-8}\;C/m$ and ${q^{2}=q_{0}^{2}=5\times 10^{-10}\;m}^{2}$, respectively. To assess the effects of the strain- and electric field-gradients, the size-factors $l^{2}$, $b_{i}$, $q^{2}$ and *f* in constitutive equations are defined by
$$l^{2}=\alpha l_{0}^{2},\;f_{i}=\alpha f_{0},\;b_{i}=\beta b_{0},\;q^{2}=\beta q_{0}^{2}.$$

To investigate influence of the strain gradient and electric intensity gradient parameters various integer numbers α and β are selected in numerical analyses.

Crack opening displacement and induced electric potential are presented in Figure 5. One can observe that both displacement and electric potential are reduced if the converse flexoelectricity is growing (larger *β*).

## 3. Gradient Theory in Thermoelectric Materials

The thermoelectric conversion efficiency is high if the thermal conductivity is low. It can be reduced significantly in nanosized structures. It is due to comparable sizes of phonon mean free path and the structure. Phonons are scattered on interfaces and thermal conductivity is reduced. For this purpose it is needed to develop a theory of heat conduction, where size effect is considered. It is well known that there is no size effect considered in the classical local theory of Fourier heat conduction. Similar to the elasticity problems in nanostructures it is possible to consider the size effect here through the nonlocal heat transport [19]. The heat flux vector in nonlocal theory is given by
(40)$$\lambda_{i}(x)=-\int_{V}\alpha (x-y)\kappa_{ij}(y)\theta_{,j}(y)dV(y),$$
where the temperature differences are denoted by $\theta =T-T_{0}$ with the reference temperature $T_{0}$, $\kappa_{ij}$ is the thermal conductivity and α(x−y) is a nonlocal kernel function.

The nonlocal weight function can be selected as
(41)$$\alpha =\frac{1}{4\pi l^{2}\rho }\exp(-\rho /l),$$
where $\rho =\left|x-y\right|$ distance and *l* is a characteristic length material parameter.

It is easy to show that weight function (41) satisfies the Helmholtz equation
(42)$$\left(1-l^{2}\nabla^{2}\right)\alpha (\left|x-y\right|=\delta (x-y),$$
where δ(x−y) is the Dirac function.

Then, the integral expression (40) is reduced to the Helmholtz equation
(43)$$\left(1-l^{2}\nabla^{2}\right)\lambda_{i}=-\kappa_{ij}\theta_{,j}\;\mathrm{or}\;\left(1-l^{2}\nabla^{2}\right)\lambda_{i,i}=w$$
where w is the volume density of heat source.

By this way it is possible to replace the integro-differential form of the constitutive law in (40) by a more convenient differential form given in (43). Then, higher-order derivatives in the governing equation appear in this non-local theory of heat conduction than in the classical local Fourier theory.

Formally, the same governing equation as given in Equation (43), can be obtained also in the higher-grade theory with including the higher-grade heat flux $m_{ik}$ (i.e., canonically conjugated fields with $\theta_{,jk}$) into constitutive equations in addition to the classical heat flux $\lambda_{i}$ as
(44)$$\lambda_{i}=-\kappa_{ij}\theta_{,j},$$
(45)$$m_{ik}=-l^{2}\kappa_{ij}\theta_{,jk}.$$

The constitutive equations for thermoelectric materials with higher order heat conduction theory can be written as
$$\lambda_{i}=-{\bar{\kappa }}_{ij}\theta_{,j}+{\bar{\zeta }}_{ij}E_{j}$$
(46)$$J_{i}=s_{ij}E_{j}-\zeta_{ij}\theta_{,j},$$
$$m_{ik}=-l^{2}\kappa_{ij}\theta_{,jk}$$
where the electric current density is denoted by $J_{i}$ and $s_{ij}$, $\zeta_{ij}$, ${\bar{\zeta }}_{ij}$ are the electrical conductivity measured with keeping uniform temperature, Seebeck and Peltier coefficients, respectively. Note that the latter two coefficients are correlated via the absolute temperature T as ${\bar{\zeta }}_{ij}=\zeta_{ij}T$, with $T=T_{0}+\theta$, where $T_{0}$ is the reference temperature. Furthermore, ${\bar{\kappa }}_{ij}=(\kappa_{ij}+\kappa_{ij}^{e})$ is composed of the heat conduction $\kappa_{ij}$ measured when $J_{i}=0$ and contribution to heat conduction $\kappa_{ij}^{e}$ because of electric current [42]. The Seebeck coefficient is proportional to the electric current conductivity $\zeta_{ij}=\alpha s_{ij}$, ${\bar{\zeta }}_{ij}=\alpha s_{ij}T$ and ${\bar{\kappa }}_{ij}=(\kappa_{ij}+\alpha^{2}s_{ij}T)$.

The electric intensity vector $E_{j}$ is related to the electric potential ϕ by
(47)$$E_{j}=-\phi_{,j}.$$

Next, orthotropic material properties are considered and the matrix form of constitutive Equations (46) for 2D problems are given by
(48)$$\left(\begin{array}{c} J_{1}\\ J_{2} \end{array}\right)=\left(\begin{array}{cc} s_{11} & 0\\ 0 & s_{22} \end{array}\right)\left(\begin{array}{c} E_{1}\\ E_{2} \end{array}\right)-\left(\begin{array}{cc} \zeta_{11} & 0\\ 0 & \zeta_{22} \end{array}\right)\left(\begin{array}{c} \theta_{,1}\\ \theta_{,2} \end{array}\right)=\left[S\right]\left\{E\right\}-\left[Z\right]\left(\begin{array}{c} \theta_{,1}\\ \theta_{,2} \end{array}\right),$$
(49)$$\left(\begin{array}{c} \lambda_{1}\\ \lambda_{2} \end{array}\right)=\left(\begin{array}{cc} {\bar{\zeta }}_{11} & 0\\ 0 & {\bar{\zeta }}_{22} \end{array}\right)\left(\begin{array}{c} E_{1}\\ E_{2} \end{array}\right)-\left(\begin{array}{cc} {\bar{\kappa }}_{11} & 0\\ 0 & {\bar{\kappa }}_{22} \end{array}\right)\left(\begin{array}{c} \theta_{,1}\\ \theta_{,2} \end{array}\right)=\left[\bar{Z}\right]\left\{E\right\}-\left[\bar{\kappa }\right]\left(\begin{array}{c} \theta_{,1}\\ \theta_{,2} \end{array}\right),$$
(50)$$\left(\begin{array}{c} m_{1k}\\ m_{2k} \end{array}\right)=-l^{2}\left(\begin{array}{cc} \kappa_{11} & 0\\ 0 & \kappa_{22} \end{array}\right)\left(\begin{array}{c} \theta_{,1k}\\ \theta_{,2k} \end{array}\right)=-l^{2}\left[\kappa \right]\left(\begin{array}{c} \theta_{,1k}\\ \theta_{,2k} \end{array}\right).$$

Then, the governing equations for stationary thermoelectric problem are given by conservation of energy and electric charge as
$$\lambda_{i,i}-m_{ik,ik}=w$$
(51)$$J_{i,i}=0.$$

The weak form of these equations can be written as
(52)$$\begin{array}{cc} \; & \int_{V}\left(J_{i}\delta \phi_{,i}+\lambda_{i}\delta \theta_{,i}+m_{ik}\delta \theta_{,ik}\right)dV+\int_{V}w\delta \theta dV=\\ \; & \;\;\;\;\;\;\;\;\;\;\;\;=-\int_{V}\left[J_{i,i}\delta \phi +\left(\lambda_{i,i}-w\right)\delta \theta +m_{ik,k}\delta \theta_{,i}\right]dV+\\ \; & \;\;\;\;\;\;\;\;\;\;\;\;\;\;\;+\int_{\partial V}\left(n_{i}J_{i}\delta \phi +n_{i}\lambda_{i}\delta \theta +n_{k}m_{ik}\delta \theta_{,i}\right)d\Gamma =\\ \; & \;\;\;\;\;\;\;\;\;\;\;\;\;\;\;-\int_{V}\left[J_{i,i}\delta \phi +\left(\lambda_{i,i}-m_{ik,ik}-w\right)\delta \theta \right]dV+\\ \; & \;\;\;\;\;\;\;\;\;\;\;\;\;\;\;+\int_{\partial V}\left[n_{i}J_{i}\delta \phi +n_{i}\left(\lambda_{i}-m_{ik,k}\right)\delta \theta +n_{k}m_{ik}\delta \theta_{,i}\right]d\Gamma =\\ \; & \;\;\;\;\;\;\;\;\;\;\;\;=-\int_{V}\left[J_{i,i}\delta \phi +\left(\lambda_{i,i}-m_{ik,ik}-w\right)\delta \theta \right]dV-\\ \; & \;\;\;\;\;\;\;\;\;\;\;\;\;\;\;\;-\int_{\partial V}\left(\Lambda \delta \theta +P\delta p+Q\delta \phi \right)d\Gamma \end{array}$$
where P, Q and Λ are independent boundary densities conjugated with p=∂θ/∂n, ϕ and θ, respectively and given as $P=n_{k}n_{i}m_{ik}$, $Q=n_{k}J_{k}$
(53)$$\Lambda =n_{j}\left(\lambda_{i}-m_{ik,k}\right)-\frac{\partial \mu }{\partial \pi }+\sum_{c}〚\mu (x^{c})〛\delta (x-x^{c})$$
(54)$$\mu =n_{k}\pi_{i}m_{ik}$$
with Λ being the heat flux, $n_{i}$ and $\pi_{i}$ are the Cartesian component of the unit tangent vector on Γ and the jump at a corner on the oriented boundary contour Γ is defined as $〚\mu (x^{c})〛:=\mu (x^{c}-)-\mu (x^{c}+)$.

The rate of work of the external “forces” $\left(\bar{\Lambda },\bar{P},\bar{Q}\right)$ and body source is given by
(55)$$\delta W=\int_{\Gamma_{\Lambda }}\bar{\Lambda }\delta \theta d\Gamma +\int_{\Gamma_{P}}\bar{P}\delta pd\Gamma +\int_{\Gamma_{Q}}\bar{Q}\delta \phi d\Gamma +\int_{V}w\delta \theta dV.$$

If only the Joule heating plays the role of heat sources,
(56)$$\delta W=\int_{\Gamma_{\Lambda }}\bar{\Lambda }\delta \theta d\Gamma +\int_{\Gamma_{P}}\bar{P}\delta pd\Gamma +\int_{\Gamma_{Q}}\bar{Q}\delta \phi d\Gamma +\int_{V}E_{i}J_{i}\delta \theta dV,$$
and the governing equations become
(57)$$\lambda_{i,i}(x)-m_{ik,ik}(x)-E_{i}(x)J_{i}(x)=0,\;J_{i,i}(x)=0$$

Furthermore, from the weak formulation, one can deduce the following boundary conditions for coupled thermoelectric problem considered within higher-grade theory

*essential b.c.*:$$\theta (x)=\bar{\theta }(x)\;\mathrm{on}\;\Gamma_{\theta },\;\Gamma_{\theta }\subset \Gamma \;p(x)=\bar{p}(x)\;\mathrm{on}\;\Gamma_{p},\;\Gamma_{p}\subset \Gamma \;\phi (x)=\bar{\phi }(x)\;\mathrm{on}\;\Gamma_{\phi },\;\Gamma_{\phi }\subset \Gamma$$

*natural b.c.*:$$\Lambda (x)=\bar{\Lambda }(x)\;\mathrm{on}\;\Gamma_{\Lambda },\;\Gamma_{\Lambda }\cup \Gamma_{\theta }=\Gamma ,\;\Gamma_{\Lambda }\cap \Gamma_{\theta }=\emptyset \;P(x)=\bar{P}(x)\;\mathrm{on}\;\Gamma_{P},\;\Gamma_{P}\cup \Gamma_{p}=\Gamma ,\;\Gamma_{P}\cap \Gamma_{p}=\emptyset \;Q(x)=\bar{Q}(x)\;\mathrm{on}\;\Gamma_{Q},\;\Gamma_{Q}\cup \Gamma_{\phi }=\Gamma ,\;\Gamma_{Q}\cap \Gamma_{\phi }=\emptyset$$

Substituting the constitutive relationships into the governing equations, we obtain the nonlinear system of the PDEs for primary field variables θ and ϕ
$$\kappa_{ij}\left(1-l^{2}\nabla^{2}\right)\theta_{,ij}+\zeta_{ij}T\left(\phi_{,ij}+\alpha \theta_{,ij}\right)+\zeta_{ij}\theta_{,i}\left(\phi_{,j}+\alpha \theta_{,j}\right)+\phi_{,i}\left(s_{ij}\phi_{,j}+\zeta_{ij}\theta_{,j}\right)=0,$$
$$s_{ij}\phi_{,ij}+\zeta_{ij}\theta_{,ij}=0.$$

Recall that owing to the Joule heat, the problem is nonlinear even if the temperature dependence of material coefficients were neglected. Finally, making use the proportionality relationship $\zeta_{ij}=\alpha s_{ij}$, the system of governing equations become
$$\kappa_{ij}\left(1-l^{2}\nabla^{2}\right)\theta_{,ij}+s_{ij}\left(\phi_{,i}+\alpha \theta_{,i}\right)\left(\phi_{,j}+\alpha \theta_{,j}\right)=0,$$
$$s_{ij}\left(\phi_{,ij}+\alpha \theta_{,ij}\right)=0.$$

### 3.1. The MLPG Formulation in Thermoelectricity

One can see in the previous chapter that MLPG method is based on the local weak-form with local fictitious subdomains $\Omega^{q}$. The local weak-form of the first governing Equation (57) is given as
(58)$$\int_{\Omega^{q}}\left[\lambda_{i,i}(x)-m_{ik,ik}(x)-E_{i}J_{i}{}_{i}\right]\;u_{}^{*}(x)\;d\Omega =0,$$
where $u_{}^{*}(x)$ is a test function.

Applying the Gauss divergence theorem to two domain integrals in (58), one can write
$$\int_{\partial \Omega^{q}}\left[\lambda_{i}(x)-m_{ik,k}(x)\right]n_{i}(x)u_{}^{*}(x)d\Gamma -\int_{\Omega^{q}}\left[\lambda_{i}(x)-m_{ik,k}(x)\right]u_{,i}^{*}(x)d\Omega -$$
(59)$$-\int_{\Omega^{q}}E_{i}(x)J_{i}(x)u_{}^{*}(x)d\Omega =0,$$
where $\partial \Omega^{q}$ is the boundary of the local subdomain $\Omega^{q}$.

The test function can be arbitrary and we have selected a Heaviside step function
(60)$$u_{}^{*}(x)=\left\{\begin{array}{c} 1\;\mathrm{at}\;x\in \Omega^{q}\cup \partial \Omega^{q}\\ 0\;\mathrm{at}\;x\notin \Omega^{q}\cup \partial \Omega^{q} \end{array}\right..$$

Then, the local weak-form (59) is transformed into the local integral equation
(61)$$\int_{L^{q}+\Gamma_{\theta }^{q}}n_{i}\left(\lambda_{i}-m_{ik,k}\right)d\Gamma +\mu (x_{\Lambda }^{f})-\mu (x_{\Lambda }^{s})-\int_{\Omega^{q}}E_{i}J_{i}d\Omega =-\int_{\Gamma_{\Lambda }^{q}}\bar{\Lambda }d\Gamma ,$$
where $x_{\Lambda }^{f}$, $x_{\Lambda }^{s}$ stand for the final and starting points on $\Gamma_{\Lambda }^{q}$ with prescribed heat flux.

The local integral equation for the second governing Equation (57) is given as
(62)$$\int_{L^{q}+\Gamma_{\phi }^{q}}n_{i}J_{i}d\Gamma =-\int_{\Gamma_{Q}^{q}}\bar{Q}d\Gamma .$$

The MLS approximation of trial functions is applied for numerical solution of the above local integral Equations (61) and (62). The temperature and electric potential are approximated by [28]
(63)$$\theta^{h}(x)=\sum_{a=1}^{n}N^{a}(x){\hat{\theta }}^{a},\;\phi^{h}(x)=\sum_{a=1}^{n}N^{a}(x){\hat{\phi }}^{a},$$
where ${\hat{\theta }}^{a}$ and ${\hat{\phi }}^{a}$ are fictitious nodal parameters for temperature and electric potential, respectively and $N^{a}(x)$ is the shape function related to the node *a*. The number of nodes *n* is explained in Section 2.2.

From the definition of heat flux $\lambda_{i}$ and the higher-grade heat flux $m_{ik}$ in (44), (45) and approximation of temperature (63) we get
(64)$$\lambda^{h}(x)={\left(\begin{array}{c} \lambda_{1}\\ \lambda_{2} \end{array}\right)}^{h}=-\bar{\kappa }\sum_{a=1}^{n}P^{a}(x){\hat{\theta }}^{a}-\bar{Z}\sum_{a=1}^{n}P^{a}(x){\hat{\phi }}^{a},\;m_{k}^{h}(x)=-l^{2}\kappa \sum_{a=1}^{n}P_{,k}^{a}(x){\hat{\theta }}^{a},$$
where
(65)$$P^{a}(x)=\left[\begin{array}{c} N_{,1}^{a}\\ N_{,2}^{a} \end{array}\right],\;P_{,k}^{a}(x)=\left[\begin{array}{c} N_{,1k}^{a}\\ N_{,2k}^{a} \end{array}\right].$$

The electric current density and intensity of electric field are approximated by
(66)$$J_{}^{h}(x)=-S\sum_{a=1}^{n}P^{a}(x){\hat{\phi }}^{a}-Z\sum_{a=1}^{n}P^{a}(x){\hat{\theta }}^{a},\;E^{T}=-\sum_{c=1}^{n}P^{cT}(x){\hat{\theta }}^{c}.$$

The incomplete heat flux $n_{i}(\lambda_{i}-m_{ik,k})$ and $\mu =n_{k}\pi_{i}m_{ik}$ are approximated as
$$n_{i}(\lambda_{i}-m_{ik,k})\approx \left(l^{2}F_{\theta }(x)\sum_{a=1}^{n}P_{,kk}^{a}(x)-{\bar{F}}_{\theta }(x)\sum_{a=1}^{n}P^{a}(x)\right){\hat{\theta }}^{a}-{\bar{F}}_{\phi }(x)\sum_{a=1}^{n}P^{a}(x){\hat{\phi }}^{a},$$
(67)$$\mu \approx -l^{2}n_{k}(x)F_{\mu }(x)\sum_{a=1}^{n}P_{,k}^{a}(x){\hat{\theta }}^{a},$$
where
$$F_{\theta }(x)=\left[\begin{array}{cc} n_{1}(x) & n_{2}(x) \end{array}\right]\left[\kappa \right]=\left[\begin{array}{cc} n_{1}(x)\kappa_{11} & n_{2}(x)\kappa_{22} \end{array}\right],$$
$${\bar{F}}_{\theta }(x)=\left[\begin{array}{cc} n_{1}(x) & n_{2}(x) \end{array}\right]\left[\bar{\kappa }(x)\right]=\left[\begin{array}{cc} n_{1}(x){\bar{\kappa }}_{11}(x) & n_{2}(x){\bar{\kappa }}_{22}(x) \end{array}\right],$$
$${\bar{F}}_{\phi }(x)=\left[\begin{array}{cc} n_{1}(x) & n_{2}(x) \end{array}\right]\left[\bar{Z}(x)\right]=\left[\begin{array}{cc} n_{1}(x){\bar{\zeta }}_{11}(x) & n_{2}(x){\bar{\zeta }}_{22}(x) \end{array}\right],$$
$$F_{\mu }(x)=\left[\begin{array}{cc} \pi_{1}(x) & \pi_{2}(x) \end{array}\right]\left[\kappa \right]=\left[\begin{array}{cc} \pi_{1}(x)\kappa_{11} & \pi_{2}(x)\kappa_{22} \end{array}\right],$$
with ${\bar{\kappa }}_{ij}(x)=\left(\kappa_{ij}+\alpha^{2}s_{ij}(T_{0}+\theta (x))\right)$, ${\bar{\zeta }}_{ij}(x)=\zeta_{ij}\left(T_{0}+\theta (x)\right)$.

The essential boundary conditions are satisfied in the strong-form at nodal points $\zeta^{b}\in \left(\Gamma_{\theta }^{b}\cup \Gamma_{p}^{b}\cup \Gamma_{\phi }^{b}\right)\subset \partial \Omega$. These conditions follows directly from Equations (63) $\sum_{a=1}^{n}N^{a}(\zeta^{b}){\hat{\theta }}^{a}=\bar{\theta }(\zeta^{b})\;\mathrm{for}\;\zeta^{b}\in \Gamma_{\theta }$, $N(\zeta^{b})\sum_{a=1}^{n}P^{a}(\zeta^{b}){\hat{\theta }}^{a}=\bar{p}(\zeta^{b})\;\mathrm{for}\;\zeta^{b}\in \Gamma_{p}$,
(68)$$\sum_{a=1}^{n}N^{a}(\zeta^{b}){\hat{\phi }}^{a}=\bar{\phi }(\zeta^{b})\;\mathrm{for}\;\zeta^{b}\in \Gamma_{\phi },$$
where $N(x)=\left[\begin{array}{cc} n_{1}(x) & n_{2}(x) \end{array}\right]$.

Substituting the MLS-approximation (64)–(67) into the local boundary-domain integral Equations (61) and (62), we obtain the nonlinear system of algebraic equations
(69)$$\begin{array}{cc} \; & \int_{L^{\;q}+\Gamma_{\theta }^{q}}\left[\left(l^{2}F_{\theta }\left(x\right)\sum_{a=1}^{n}P_{,kk}^{a}\left(x\right)-{\bar{F}}_{\theta }\left(x\right)\sum_{a=1}^{n}P^{a}\left(x\right)\right){\hat{\theta }}^{a}-{\bar{F}}_{\phi }\left(x\right)\sum_{a=1}^{n}P^{a}\left(x\right){\hat{\phi }}^{a}\right]d\Gamma +\\ \; & \;\;\;\;\;\;\;\;\;\;\;\;\;\;\;+l^{2}\left(n_{k}\left(x_{\Lambda }^{s}\right)F_{\mu }\left(x_{\Lambda }^{s}\right)\sum_{a=1}^{n}P_{,k}^{a}\left(x_{\Lambda }^{s}\right)-n_{k}\left(x_{\Lambda }^{f}\right)F_{\mu }\left(x_{\Lambda }^{f}\right)\sum_{a=1}^{n}P_{,k}^{a}\left(x_{\Lambda }^{f}\right)\right){\hat{\theta }}^{a}-\\ \; & \;\;\;\;\;\;\;\;\;\;\;\;\;\;\;-\int_{\Omega^{q}}\sum_{c=1}^{n}{\hat{\theta }}^{\;c}P^{\;cT}\left(x\right)\left(S\sum_{a=1}^{n}P^{a}\left(x\right){\hat{\phi }}^{a}+Z\sum_{a=1}^{n}P^{a}\left(x\right){\hat{\theta }}^{a}\right)d\Omega =-\int_{\Gamma_{\Lambda }^{q}}\bar{\Lambda }d\Gamma \end{array}$$
(70)$$\int_{L^{q}+\Gamma_{\phi }^{q}}N(x)\left(S\sum_{a=1}^{n}P^{a}(x){\hat{\phi }}^{a}+Z\sum_{a=1}^{n}P^{a}(x){\hat{\theta }}^{a}\right)d\Gamma =\int_{\Gamma_{Q}^{q}}\bar{Q}d\Gamma .$$

This system can be solved iteratively as
(71)$$\begin{array}{cc} \; & \int_{L^{\;q}+\Gamma_{\theta }^{q}}\left[\left(l^{2}F_{\theta }\left(x\right)\sum_{a=1}^{n}P_{,kk}^{a}\left(x\right)-{\bar{F}}_{\theta }^{\left(k-1\right)}\left(x\right)\sum_{a=1}^{n}P^{a}\left(x\right)\right){\hat{\theta }}^{a\;(k)}-{\bar{F}}_{\phi }^{\left(k-1\right)}\left(x\right)\sum_{a=1}^{n}P^{a}\left(x\right){\hat{\phi }}^{a\;(k)}\right]d\Gamma +\\ \; & \;\;\;\;\;\;\;\;\;\;\;\;\;\;\;\;\;+l^{2}\left(n_{k}\left(x_{\Lambda }^{s}\right)F_{\mu }\left(x_{\Lambda }^{s}\right)\sum_{a=1}^{n}P_{,k}^{a}\left(x_{\Lambda }^{s}\right)-n_{k}\left(x_{\Lambda }^{f}\right)F_{\mu }\left(x_{\Lambda }^{f}\right)\sum_{a=1}^{n}P_{,k}^{a}\left(x_{\Lambda }^{f}\right)\right){\hat{\theta }}^{a\;(k)}-\\ \; & \;\;\;\;\;\;\;\;\;\;\;\;\;\;\;\;\;-\int_{\Omega^{q}}\sum_{c=1}^{n}{\hat{\theta }}^{c\;(k-1)}P^{\;cT}\left(x\right)\left(S\sum_{a=1}^{n}P^{a}\left(x\right){\hat{\phi }}^{a\;(k)}+Z\sum_{a=1}^{n}P^{a}\left(x\right){\hat{\theta }}^{a\;(k)}\right)d\Omega =-\int_{\Gamma_{\Lambda }^{q}}\bar{\Lambda }d\Gamma \end{array}$$
(72)$$\int_{L^{q}+\Gamma_{\phi }^{q}}N(x)\left(S\sum_{a=1}^{n}P^{a}(x){\hat{\phi }}^{a(k)}+Z\sum_{a=1}^{n}P^{a}(x){\hat{\theta }}^{a(k)}\right)d\Gamma =\int_{\Gamma_{Q}^{q}}\bar{Q}d\Gamma ,$$
with (k=1,2,…), ${\hat{\theta }}^{a(0)}=0$, ${\hat{\phi }}^{a\;(0)}=0$ and
$${\bar{F}}_{\theta }^{\left(k-1\right)}(x)=\left[\begin{array}{cc} n_{1}(x){\bar{\kappa }}_{11}^{\left(k-1\right)}(x) & n_{2}(x){\bar{\kappa }}_{22}^{\left(k-1\right)}(x) \end{array}\right],$$
$${\bar{F}}_{\phi }^{\left(k-1\right)}(x)=\left[\begin{array}{cc} n_{1}(x){\bar{\zeta }}_{11}^{\left(k-1\right)}(x) & n_{2}(x){\bar{\zeta }}_{22}^{\left(k-1\right)}(x) \end{array}\right],$$
where ${\bar{\kappa }}_{ij}^{\left(k-1\right)}(x)=\left(\kappa_{ij}+\alpha^{2}s_{ij}(T_{0}+\theta^{\left(k-1\right)}(x))\right)$, ${\bar{\zeta }}_{ij}^{\left(k-1\right)}(x)=\zeta_{ij}\left(T_{0}+\theta^{\left(k-1\right)}(x)\right)$.

### 3.2. Numerical Examples

An axially symmetric thermoelectric problems, as shown in Figure 6, is analyzed in the example. The thermoelectric material Bi_2_Te_3_, is considered. It has the following material constants [42] with isotropic properties:(73)$$s=1.1\times 10^{5}\;\mathrm{Am}/V,\;\alpha =\zeta /s=2\times 10^{-4}\;V^{2}/KAm,\;\kappa =1.6\;W/Km.$$

Characteristic length for the selected material structure is $l=5\times 10^{-9}\;m$.

Following geometry of the hollow cylinder is considered: internal radius *r*_1_ = 1 × 10^−7^ m and external radius *r*_2_ = 2.5 × 10^−7^ m. Vanishing values of electric potentials are prescribed on inner and external surfaces. Furthermore, vanishing values of the temperature gradients on both surfaces are considered.

The coupled thermo-electric problem is analyzed numerically. The influence of the tube thickness $L=r_{2}-r_{1}$ on temperature and induced electric potential is investigated. Numerical results are presented in Figure 7 and Figure 8. The induced electric potential grows with increasing the value *l/L*. In classical thermoelectricity, it is possible to find the analytical solution.

## 4. Conclusions

The meshless Petrov–Galerkin (MLPG) method was successfully applied to multiphysical problems described by advanced continuum models with microstructural effects. Strain- and electric intensity vector-gradients are considered in constitutive equations for electric displacement and stresses in flexoelectricity, respectively. Similarly, the constitutive equations for thermoelectric materials contain higher-order derivatives of temperature in the higher-grade heat conduction theory. It allows to describe the heat transfer in nanostructures more realistic. The governing equations are derived for both multiphysical problems, where size effects are considered. These equations contain higher order of derivatives of physical fields than in the classical continuum models. Application of classical domain discretization methods to corresponding boundary value problems brings some difficulties with continuity of approximated fields.

The proposed MLPG computational method with the MLS approximation of fields is very convenient to solve governing equations of gradient theory with high-order derivatives. The order of continuity of the MLS approximation can be tuned to a desired value very easily. Therefore, the present computational method is promising to be applied to multiphysical problems described by gradient theories. The operation of the developed computational scheme is verified via comparison of partial numerical results with analytical solutions and/or FEM results.

## Figures and Tables

**Figure 1 materials-13-02527-f001:**
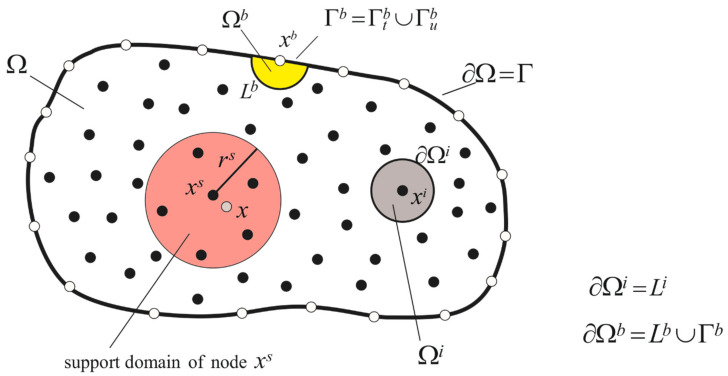
Local subdomains $\Omega^{i}$ and $\Omega^{b}$ with their boundaries for moving least-squares (MLS) approximation of the trial function; support domain of weight function around node $x^{s}$.

**Figure 2 materials-13-02527-f002:**
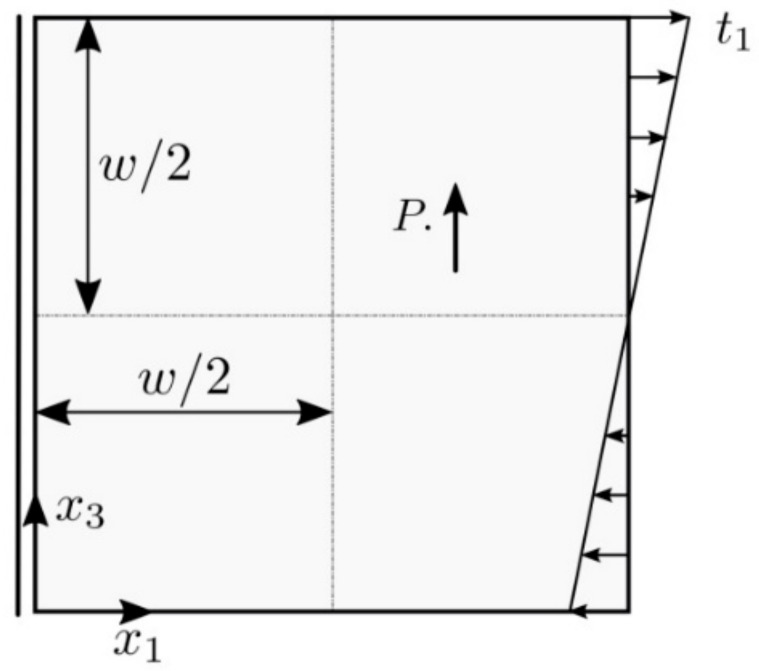
A square piezoelectric panel under bending.

**Figure 3 materials-13-02527-f003:**
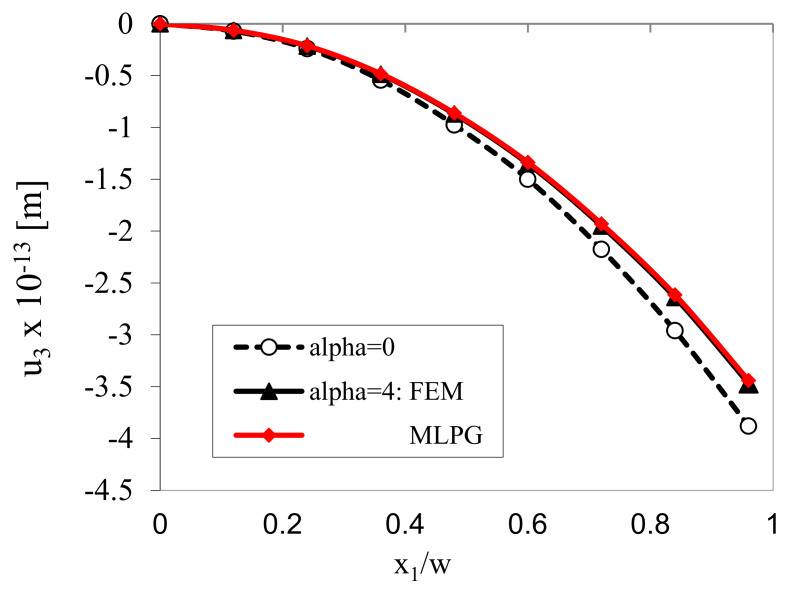
Variation of the mechanical displacement *u*_3_ at *x*_3_ = *w*/2.

**Figure 4 materials-13-02527-f004:**
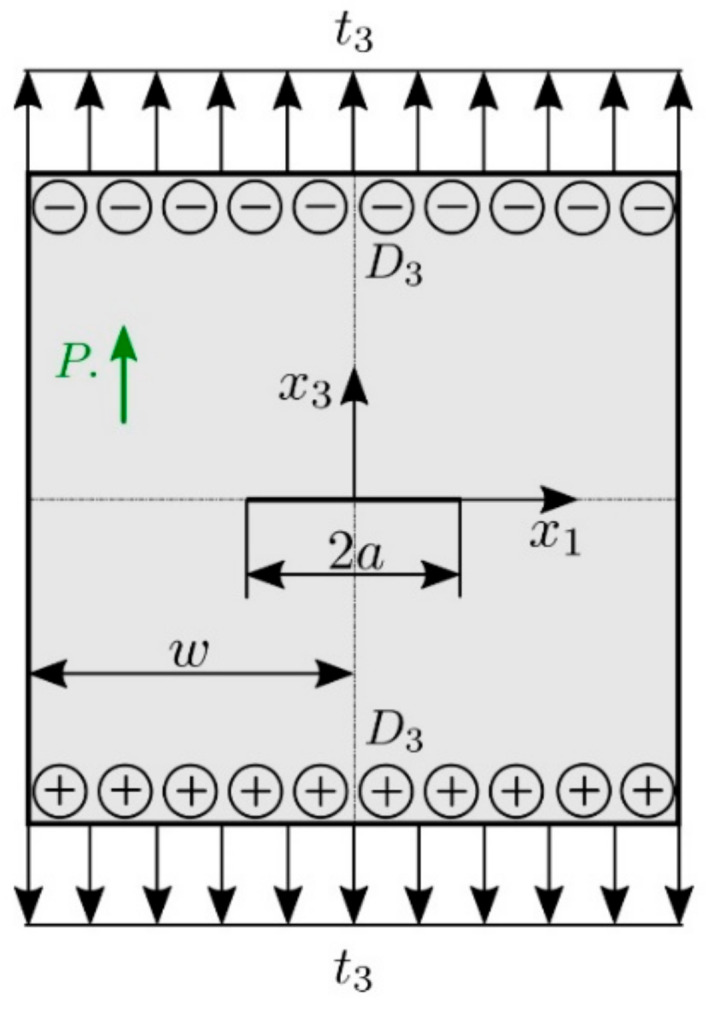
Crack in a square plate under an electro-mechanical loading.

**Figure 5 materials-13-02527-f005:**
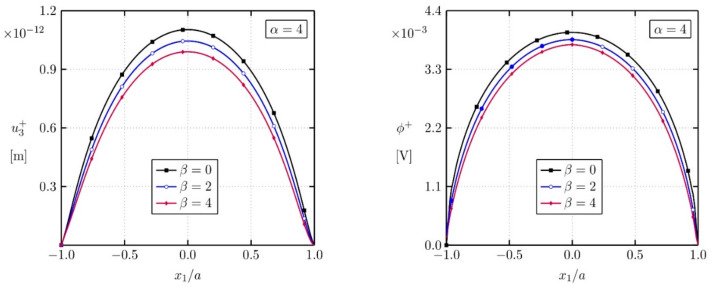
Crack opening-displacements $u_{3}^{+}$ and electrical potentials *φ*^+^ of the upper crack-face for different factors *β.*

**Figure 6 materials-13-02527-f006:**
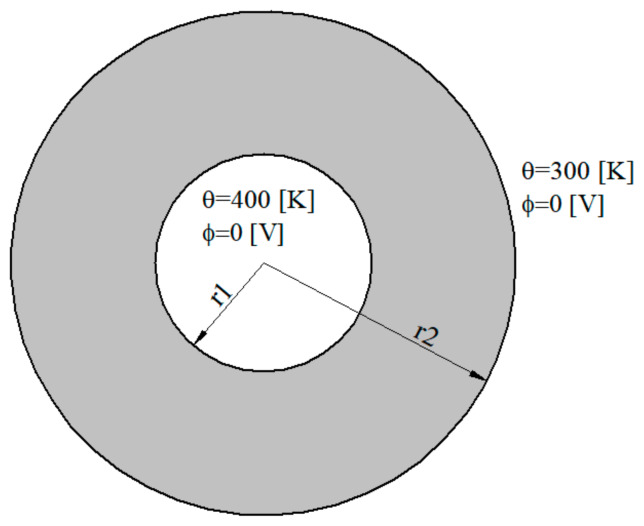
Geometry and boundary conditions.

**Figure 7 materials-13-02527-f007:**
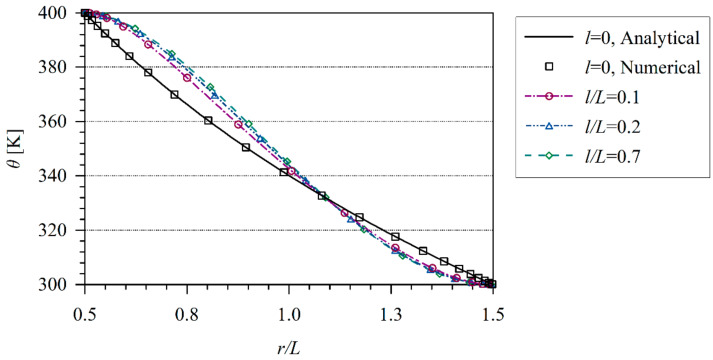
Temperature variation along non dimensional *x/L* coordinate in hollow cylinder.

**Figure 8 materials-13-02527-f008:**
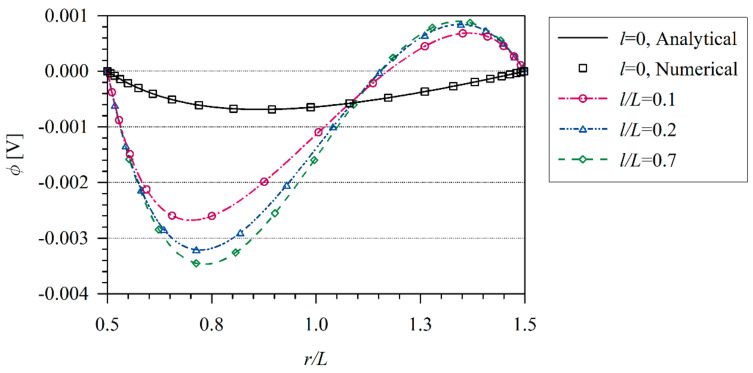
Electric potential variation along non dimensional *x/L* coordinate in hollow cylinder.

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
