# Peer review of "The Meshless Analysis of Scale-Dependent Problems for Coupled Fields"

_materials, 2020, doi:10.3390/ma13112527_

Round 1

Reviewer 1 Report

The authors of the manuscript “The meshless analysis of scale dependent problems 2 for coupled fields” propose the meshless local Petrov-Galerkin (MLPG) method for the solution of multiphysical problems in flexoelectricity and thermoelectricity. The mathematical formulation is well documented and extensive. Several benchmark problems are solved to demonstrate the accuracy of the method. Comparisons are performed with solutions obtained by the Finite Element method. While the presented work is of interest, I feel that the paper is not publishable at its current form. The structure is quite confusing. Theory and numerical examples are mixed throughout the manuscript. The authors present the application of MLPG in two different physical problems. Probably it could be more proper to present each one of the physical problems in different papers. The quality of English is very low, and the introduction is confusing. It requires significant improvements. Below I present few comments to improve the work.

Comments

Equations 21, 22 are confusing. It doesn’t make sense to me adding and substracting 0 from the same variable. Could you explain what is 0 in your paper? Is it 0-, 0+ ?

Figure 1 should mention in the legend the field nodes xq and xb which are present in the figure

In equation 30, 31, 33, 34 why you use the index 3 and not the index 2?

Grammar & Syntax

-------------------------

Abstract

The variational principle is applied to derive the governing equations considered constitutive equations - Bad syntax, too confusing

The order of derivatives in governing equations is higher than in equations obtained from classical theory  à The order of derivatives in the governing equations is higher than in equations obtained from classical theory

Applying the spatial approximations in local integral equations a system of algebraic is obtained for the nodal unknowns à Applying the spatial approximations in local integral equations, a system of algebraic equations is obtained for the nodal unknowns

Introduction

line 22 – remove Then,

line 31 strain- → strain

line 47 → simplified the former

Section 2

line 76 – remove and

line 193 → are approximated

Author Response

Reviewer #1

The authors of the manuscript “The meshless analysis of scale dependent problems 2 for coupled fields” propose the meshless local Petrov-Galerkin (MLPG) method for the solution of multiphysical problems in flexoelectricity and thermoelectricity. The mathematical formulation is well documented and extensive. Several benchmark problems are solved to demonstrate the accuracy of the method. Comparisons are performed with solutions obtained by the Finite Element method. While the presented work is of interest, I feel that the paper is not publishable at its current form. The structure is quite confusing. Theory and numerical examples are mixed throughout the manuscript. The authors present the application of MLPG in two different physical problems. Probably it could be more proper to present each one of the physical problems in different papers. The quality of English is very low, and the introduction is confusing. It requires significant improvements. Below I present few comments to improve the work.

 Authors reply: Many thanks for the Reviewer’s comment. Both flexoelectricity and thermoelectricity problems are similar from mathematical point of view to apply the MLPG. Purpose of the paper is to verify applicability of the MLPG to problems described by gradient theories with higher order derivatives with respect to classical theories. Therefore, we prefer to analyse both problems in one paper. Introduction is rewritten. Some new parts are supplied. English is improved. We decided to publish both physically different topics in one paper owing to avoiding self-plagiarism in case of two papers with similar numerical treatment. The authors believe that it is appropriate to complete the section devoted to flexoelectricity by related numerical examples before opening the discussion about the thermoelectricity finished by related illustrative numerical examples.

Comments

 Equations 21, 22 are confusing. It doesn’t make sense to me adding and substracting 0 from the same variable. Could you explain what is 0 in your paper? Is it 0-, 0+ ?

Figure 1 should mention in the legend the field nodes xq and xb which are present in the figure

In equation 30, 31, 33, 34 why you use the index 3 and not the index 2

Authors reply: Many thanks for the Reviewer’s comment. We have modified the definition of jump terms at a corner point on oriented boundary contour as

The plane-deformation problems for crystals of hexagonal symmetry (class) with being the 6-order symmetry axis. It is assumed as well as the independence of the field quantities on , i.e. , we have (Patron and Kudryavtsev 1988).

It is standard notation for 2-d problems in piezoelectricity with poling direction along the -axis.

Grammar & Syntax

-------------------------

Abstract

The variational principle is applied to derive the governing equations considered constitutive equations - Bad syntax, too confusing

The order of derivatives in governing equations is higher than in equations obtained from classical theory  à The order of derivatives in the governing equations is higher than in equations obtained from classical theory

Applying the spatial approximations in local integral equations a system of algebraic is obtained for the nodal unknowns à Applying the spatial approximations in local integral equations, a system of algebraic equations is obtained for the nodal unknowns

Introduction

line 22 – remove Then,

line 31 strain- → strain

line 47 → simplified the former

Section 2

line 76 – remove and

line 193 → are approximated

Authors reply: Many thanks for the Reviewer’s comment. All grammar & syntax errors are corrected.

Reviewer 2 Report

The manuscript proposes a MLPG scheme for the solution of 2D problems of flexoelectricity and thermoelasticity.

The manuscript is interesting, within the aim and scope of the journal, and very well-written. However, before recommending the paper for publication the authors may address the following minor and major points:

Minors:

  1. The authors have justified the application of MLPG for handling c1 problems. It is still not that clear for the reader the justification of using a meshfree approach for these problems, while FEM is still providing suitable results. Even the authors have checked the accuracy of their results with that of a FEM scheme considering the length scale. It would be beneficial to write a minimum of two paragraphs about the overview of meshfree approaches and their advantages over mesh-based approaches such as FEM.
  2. A recent category of studies in the solution of nonlocal problems are models based on the peridynamic theory (Journal of the Mechanics and Physics of Solids. 2000 Jan 1;48(1):175-209.). The majority of these models make use of strong-form discretization (Computers & structures. 2005 Jun 1;83(17-18):1526-35.). There are also a trend of active studies to design meshless methods that take the advantages of the classical and nonlocal theories in simultaneously in a meshless framework (International Journal of Mechanical Sciences. 2016 Dec 1;119:419-31.) & (Computational Mechanics. 2015 Feb 1;55(2):287-302.) && (Engineering Computations
  3. 34 No. 5, 2017 pp. 1334-1366). It would not be a bad idea to discuss on that in the introduction.

Majors:

  1. What are the limitations of the proposed approach? How much large could you take \alpha in the simulation? Can the method recover the solution of a MD model?
  2. Does the developed scheme recover the classical solution by taking \alpha equal to zero?
  3. Would it be possible to discuss the convergence rate of the proposed scheme for a problem in which the analytical solution is known? In this way, the authors may take \alpha a constant value and refine the discretized domain using smaller grid sizes.  

Author Response

Reviewer #2

Comments and Suggestions for Authors

The manuscript proposes a MLPG scheme for the solution of 2D problems of flexoelectricity and thermoelasticity.

The manuscript is interesting, within the aim and scope of the journal, and very well-written. However, before recommending the paper for publication the authors may address the following minor and major points:

Minors:

  1. The authors have justified the application of MLPG for handling c1 problems. It is still not that clear for the reader the justification of using a meshfree approach for these problems, while FEM is still providing suitable results. Even the authors have checked the accuracy of their results with that of a FEM scheme considering the length scale. It would be beneficial to write a minimum of two paragraphs about the overview of meshfree approaches and their advantages over mesh-based approaches such as FEM.

Authors reply: Many thanks for the Reviewer’s comment. The revised parts in Introduction are marked by red colour in the revised manuscript. Higher order derivatives in governing equations require the C1-continuous elements in numerical domain discretization methods to guarantee the continuity of variables and their derivatives on interfaces of elements. It is a difficult task to obey such a requirement, though certain special elements have been developed. It is more convenient to develop the mixed formulation in the finite element method (FEM) [20, 21]. The derivatives as independent variables in the mixed FEM require only C0 continuity. However, numerous degrees of freedom (DOFs) are used in each element, which make it prohibitively expensive for practical use. The order of continuity of the Moving Least-square (MLS) approximation in the Meshless Local Petrov-Galerkin method (MLPG) can be tuned to a desired value very easily [22-24]. Therefore, it is a natural ambition to apply the MLPG with MLS approximation to problems described by gradient theories with higher order derivatives in governing equations.

  1. A recent category of studies in the solution of nonlocal problems are models based on the peridynamic theory (Journal of the Mechanics and Physics of Solids. 2000 Jan 1;48(1):175-209.). The majority of these models make use of strong-form discretization (Computers & structures. 2005 Jun 1;83(17-18):1526-35.). There are also a trend of active studies to design meshless methods that take the advantages of the classical and nonlocal theories in simultaneously in a meshless framework (International Journal of Mechanical Sciences. 2016 Dec 1;119:419-31.) & (Computational Mechanics. 2015 Feb 1;55(2):287-302.) && (Engineering Computations 34 No. 5, 2017 pp. 1334-1366). It would not be a bad idea to discuss on that in the introduction.

Authors reply: Many thanks for the Reviewer’s comment. A discussion about peridynamic theory is included into Introduction and new references are supplied.

Majors:

  1. What are the limitations of the proposed approach? How much large could you take \alpha in the simulation? Can the method recover the solution of a MD model?
  2. Does the developed scheme recover the classical solution by taking \alpha equal to zero?
  3. Would it be possible to discuss the convergence rate of the proposed scheme for a problem in which the analytical solution is known? In this way, the authors may take \alpha a constant value and refine the discretized domain using smaller grid sizes.  

Authors reply: Many thanks for the Reviewer’s comment. Micro-length scale parameter is a material coefficient occurring in the phenomenological higher-grade continuum theory. Since it is difficult to get experimental data for such a parameter, we consider the parametric study of influence on response via numerical simulations using various values of this parameter. Only the values comparable with a characteristic length of microstructure are physically meaningful.   The MD calculations are not applicable to macro-structural elements from two reasons: (i) memory limitations of computer facilities; (ii) the output of MD calculations is not convenient for description of global behaviour of macro-structural elements. Recently, we have applied the MD method to solve flexoelectric problem to get unknown flexoelectric coefficients. The manuscript is accepted for publication, J. Sladek, S. Hocker, H. Lipp, V. Sladek, Q. Deng: Atomistic approach for the evaluation of direct flexoelectric coefficients in gradient theory, Ferroelectrics, Vol 569 (2020).

  1. The classical and higher-grade problems are to be solved individually, since the governing equations in the higher-grade model are given by the PDE of higher order than in classical theory, and moreover, some additional boundary conditions are required.

3.Convergence rate of the MLPG with MLS approximations has been investigated in many earlier papers. Since the method is quite universal, the values of numerical model parameters found form sensitivity analyses in numerical experiments in the past are applicable to a wide variety of boundary value problems for PDE. To keep the length of the paper reasonable, we avoided the convergence analyses.

S.N. Atluri, J. Sladek, V. Sladek and T. Zhu: The local boundary integral equation (LBIE) and its meshless implementation for linear elasticity, Computational Mechanics 25, 2000, 180-198.

  1. Sladek, J. Sladek, Ch. Zhang: Computation of stresses in non-homogeneous elastic solids by local integral equation method: a comparative study, Computational Mechanics 41 (2008), 827-845.

Reviewer 3 Report

The reviewed paper presents the application of the meshless Petrov-Galerkin method for numerical analysis of selected coupled and scale dependent two-dimensional problems, including the flexoelectricity and thermoelectricity problems. Authors show the mathematical formulation of the considered problem, which is followed by the description of the numerical model. Two numerical examples are presented. Afterwards, the gradient theory, providing the size effect problem is discussed, followed by other numerical examples. The paper is briefly concluded.

Though the reviewed paper is interesting and worth publishing, numerous major remarks came to my mind, while reading it. They are listed below, in order of appearance in the text. I dare to ask the authors to give the comprehensive answer to them and prepare the revised version of their paper, per chance with enhanced readability.

major remarks (referring to row number in the reviewed manuscript)

  1. authors mentioned within the Introduction, and later in subsection 2.2., that due to C1 continuity requirements, they are using MLPG method; however, why not using well-established FEM framework with finite elements of C1 class? (plate-like ones)? Authors should give at least one reason and show at least one comparison example (MLPG vs FEM), showing clearly, why their method is more effective here. The problems with discontinuities on elements’ interfaces, mentioned later, is not a convincing reason, as they may come from numerical errors and the same may be claimed here, in case of MLPG framework. Moreover, the results for the first example do not show any advantages of MLPG over FEM.
  2. authors dealt with similar problems before, for instance, for plates and crack opening problems; to avoid raising questions concerning a self-plagiarism, authors are kindly asked to highlight the novel aspects of their work, presented here, that were not presented before.
  3. 111 – what is “e15”? in 111 – particular e have two indices, whereas in its definition/formulation (?) (112) e has three indices, again depending on e15 – what is this variable here? What is its connection with piezoelectric coefficients, mentioned earlier?
  4. the same remark is related to b – it is unclear how to determine b1, b2 and b3.
  5. 166 – in Fig.1, the thee parts mentioned in the text, are not visible (I assume the grey subdomain, assigned to xq, is considered); hence Lq, gamma^q_t and gamma^q_u are not properly explained. Reader may suspect what they are, however it should not work that way.
  6. the source and the idea behind the final and starting points in (26) is not clear,
  7. the relation between the number of nodes (n) for local MLS approximation and the size of a support is not clear; usually, the number of nodes should follow the order of differential operator in governing equations/variational principle and for MLS approximation, it is usually assumed between the lowest number required for this order and the number corresponding the order+1; as I do not see such a relation here, I would like to ask authors for explanations,
  8. applying spline-like approximation type for internal nodes, with complete circular supports, could be performed without large difficulties; however, applying (29) for cut boundary subdomains seems to be problematic – the way the size of such support is compiled in (29) is unclear.
  9. the shape functions do not have 0-1 conditions – are they really shape functions or rather pseudo-shape functions? In that case, how the essential boundary conditions are fulfilled?
  10. None is said about the numerical integration schemes, which are involved for interior and boundary of nodal subdomains. Again, the problem is integration on cut boundary subdomains seems to be omitted here.
  11. As expected, authors are considering only simple geometries like rectangle and circle with circular void. They do not seem to be justifying the selection of the meshless method, which main advantage is that it works well (better than FEM) with irregular clouds of nodes, generated for complex geometries. However in this paper, none is said about the numerical model in all numerical examples, for instance the number of nodes or the size of local supports /the number of nodes for local MLS. Moreover, none of the considered examples require generation of highly irregular cloud of nodes. Hence, my impression is that authors are using simple regular meshes, which cannot truly justified their work. Therefore, I ask authors to:
    • give brief information concerning the numerical models they have assumed for all examples,
    • show at least one example with either complex geometry and/or highly irregular mesh (per chance, generated from regular one, with random distortion of nodes locations), with a simple convergence test (e.g. for mechanical displacement), showing on a graph, in logarithmic or semi-logarithmic scale: number of nodes vs. estimated error (e.g. difference of u between neighbouring meshes) – this is the most common approach to show the convergence of the method without any mathematical proof.
  12. Discuss briefly the sensitivity of the entire model for selection of material and/or numerical parameters – on what aspects the potential users of your approach should pay attention at?

minor remark (referring to row number in the reviewed manuscript)

  • 76 – “and” seems to be obsolete

Author Response

Reviewer #3

Comments and Suggestions for Authors

The reviewed paper presents the application of the meshless Petrov-Galerkin method for numerical analysis of selected coupled and scale dependent two-dimensional problems, including the flexoelectricity and thermoelectricity problems. Authors show the mathematical formulation of the considered problem, which is followed by the description of the numerical model. Two numerical examples are presented. Afterwards, the gradient theory, providing the size effect problem is discussed, followed by other numerical examples. The paper is briefly concluded.

Though the reviewed paper is interesting and worth publishing, numerous major remarks came to my mind, while reading it. They are listed below, in order of appearance in the text. I dare to ask the authors to give the comprehensive answer to them and prepare the revised version of their paper, per chance with enhanced readability.

major remarks (referring to row number in the reviewed manuscript)

  • authors mentioned within the Introduction, and later in subsection 2.2., that due to C1 continuity requirements, they are using MLPG method; however, why not using well-established FEM framework with finite elements of C1 class? (plate-like ones)? Authors should give at least one reason and show at least one comparison example (MLPG vs FEM), showing clearly, why their method is more effective here. The problems with discontinuities on elements’ interfaces, mentioned later, is not a convincing reason, as they may come from numerical errors and the same may be claimed here, in case of MLPG framework. Moreover, the results for the first example do not show any advantages of MLPG over FEM.

Authors reply: Many thanks for the Reviewer’s comment. The main purpose of the present paper is to give the mathematical formulation of two physically different classes of coupled field problems by the MLPG method and to present some numerical simulations for size effects within the higher-grade continuum theory. There is made also a comparison with FEM calculations, but the complete comparative analysis is not the aim of this paper. The study of computational stability (accuracy, convergence) and efficiency are not prohibited and can be done in future. It is well known that development of C1-continuous elements in FEM is extremely difficult task, though certain special elements have been developed. Therefore, there are some efforts to develop a mixed formulation in the FEM, where derivatives are approximated as independent variables with C0 continuity. However, numerous degrees of freedom (DOFs) are used in each element, which make it prohibitively expensive for practical use. In the Moving Least-square (MLS) approximation the continuity can be tuned to a desired value very easily. It was a main motivation why we have made attempt to develop the MLPG with MLS for considered problems with higher derivatives in governing equations.

  • authors dealt with similar problems before, for instance, for plates and crack opening problems; to avoid raising questions concerning a self-plagiarism, authors are kindly asked to highlight the novel aspects of their work, presented here, that were not presented before.

Authors reply: We have never applied the MLPG to converse flexoelectricity and thermoelectricity. To the best of the authors’ knowledge, no work has been carried out so far to develop the MLPG for these problems. The context of the manuscript is completely novel.

  • 111 – what is “e15”? in 111 – particular e have two indices, whereas in its definition/formulation (?) (112) e has three indices, again depending on e15 – what is this variable here? What is its connection with piezoelectric coefficients, mentioned earlier?

Authors reply: Many thanks for the Reviewer’s comment. Note that , and is the standard Voigt notation for independent piezoelectric coefficients in crystals with considered symmetry properties, when the tensor of piezoelectric coefficients is defined according Eq. (8). (Patron and Kudryavtsev 1988).

  • the same remark is related to b – it is unclear how to determine b1, b2 and b3.

Authors reply: Many thanks for the Reviewer’s comment. Note that , and are three independent material coefficients occurring in the definition of the converse flexoelectric tensor , but are not components of a vector, similar as , in case of direct flexoelectricity tensor.

  • 166 – in Fig.1, the three parts mentioned in the text, are not visible (I assume the grey subdomain, assigned to xq, is considered); hence Lq, gamma^q_t and gamma^q_u are not properly explained. Reader may suspect what they are, however it should not work that way.

Authors reply: Many thanks for the Reviewer’s comment. In general, and , . It can be seen from Fig. 1, provided that is an interior node , while with if is a nodal point on the boundary contour .

  • the source and the idea behind the final and starting points in (26) is not clear,

Authors reply: Many thanks for the Reviewer’s comment. Final and starting points are occurred on the global boundary, where local boundary intersects the global boundary.

  • the relation between the number of nodes (n) for local MLS approximation and the size of a support is not clear; usually, the number of nodes should follow the order of differential operator in governing equations/variational principle and for MLS approximation, it is usually assumed between the lowest number required for this order and the number corresponding the order+1; as I do not see such a relation here, I would like to ask authors for explanations,

Authors reply: Many thanks for the Reviewer’s comment. The number of nodes, n, used for the approximation is determined by the size of support domain of the weight functionof each node . The node is involved into the approximation at the sample point, only if lies within the support domain of the node , i.e., if . A necessary condition for a regular MLS approximation is that at least weight functions are non-zero (i.e. ) for each sample point , where is order of the complete basis functions used in MLS approximation. A sufficient number of nodes must be involved in order to ensure the regularity of evaluation of shape functions [27]. A small size of the support domains may induce larger oscillations in the nodal shape functions.

  • applying spline-like approximation type for internal nodes, with complete circular supports, could be performed without large difficulties; however, applying (29) for cut boundary subdomains seems to be problematic – the way the size of such support is compiled in (29) is unclear.

Authors reply: Many thanks for the Reviewer’s comment. The circular shape of subdomain enables us to utilize polar coordinates and facilitate the integrations. Since the radius of local subdomain is rather small, the subdomain around a boundary node can be considered as a section of complete circular domain and is a circular arc (Fig.1). The approximation at any sample point is supported by each node whose support domain involve the sample point . If the number of nodes supporting the approximation at were smaller than a minimum required number (say because of location of sample point close to the boundary of analysed domain), additional closest nodes would be supplemented as supporting nodes in the adopted numerical model.

  • the shape functions do not have 0-1 conditions – are they really shape functions or rather pseudo-shape functions? In that case, how the essential boundary conditions are fulfilled?

Authors reply: Many thanks for the Reviewer’s comment. In the MLS approximation, the shape functions do not satisfy the Kronecker -property. Therefore the nodal unknowns,are not nodal values of displacements and electric potential, but these nodal unknowns can be used for approximation of real nodal values at an arbitrary nodal point by utilizing (28) at . Thus, the essential boundary conditions at nodal points can be satisfied in the strong-form, as it is written in eq. (36).

  • None is said about the numerical integration schemes, which are involved for interior and boundary of nodal subdomains. Again, the problem is integration on cut boundary subdomains seems to be omitted here.

Authors reply: Many thanks for the Reviewer’s comment. The Gaussian quadrature is used for integrations with using polar coordinate system; the subdomain is very small and it can be selected as a section of circular domain in case of cut boundary subdomain.

  • As expected, authors are considering only simple geometries like rectangle and circle with circular void. They do not seem to be justifying the selection of the meshless method, which main advantage is that it works well (better than FEM) with irregular clouds of nodes, generated for complex geometries. However in this paper, none is said about the numerical model in all numerical examples, for instance the number of nodes or the size of local supports /the number of nodes for local MLS. Moreover, none of the considered examples require generation of highly irregular cloud of nodes. Hence, my impression is that authors are using simple regular meshes, which cannot truly justified their work. Therefore, I ask authors to:
  • give brief information concerning the numerical models they have assumed for all examples,
  • show at least one example with either complex geometry and/or highly irregular mesh (per chance, generated from regular one, with random distortion of nodes locations), with a simple convergence test (e.g. for mechanical displacement), showing on a graph, in logarithmic or semi-logarithmic scale: number of nodes vs. estimated error (e.g. difference of u between neighbouring meshes) – this is the most common approach to show the convergence of the method without any mathematical proof.
  • Discuss briefly the sensitivity of the entire model for selection of material and/or numerical parameters – on what aspects the potential users of your approach should pay attention at?

minor remark (referring to row number in the reviewed manuscript)

  • 76 – “and” seems to be obsolete

Authors reply: Many thanks for the Reviewer’s comment. Information about the employed numerical model is supplemented in the revised version of the manuscript. In literature, it is lot of papers where the numerical stability (accuracy, convergence) and computational efficiency of the MLPG with MLS approximation is tested thoroughly in classical elasticity or potential problems. Sorry, it is not the purpose of this paper to focus on such thorough testing. We have only developed the MLPG formulation for the gradient theory of flexoelectricity and thermoelectricity, now. The operation of the developed computational scheme is verified via comparison of partial numerical results with analytical solutions and/or FEM results. We have only submitted the first attempt to solve thermoelectricity by the FEM in another paper. It is not standard task to solve above problems by the FEM.

Round 2

Reviewer 2 Report

The authors have clearly addressed my questions. I recommend the paper for publication in the present form. 

Reviewer 3 Report

Authors have answered all my concerns and provided appropriate corrections. Though there are some issues left that may require further discussion (like MLS shape functions, which may satisfy Kronecker delta property, providing the singular weight functions are assumed), my recommendation is to accept the manuscript in its present form.